# Human B cell lineages associated with germinal centers following influenza vaccination are measurably evolving

Kenneth B Hoehn[1], Jackson S Turner[2], Frederick I Miller[3], Ruoyi Jiang[4], Oliver G Pybus[5], Ali H Ellebedy[2,6], Steven H Kleinstein[1,4,7]*

[1]Department of Pathology, Yale School of Medicine, New Haven, United States; [2]Department of Pathology and Immunology, Washington University School of Medicine, St Louis, United States; [3]Worcester Polytechnic Institute, Worcester, United States; [4]Department of Immunobiology, Yale School of Medicine, New Haven, United States; [5]Department of Zoology, University of Oxford, Oxford, United Kingdom; [6]The Andrew M. and Jane M. Bursky Center for Human Immunology and Immunotherapy Programs, Washington University School of Medicine, St Louis, United States; [7]Interdepartmental Program in Computational Biology & Bioinformatics, Yale University, New Haven, United States

*For correspondence:
steven.kleinstein@yale.edu

**Abstract** The poor efficacy of seasonal influenza virus vaccines is often attributed to pre-existing immunity interfering with the persistence and maturation of vaccine-induced B cell responses. We previously showed that a subset of vaccine-induced B cell lineages are recruited into germinal centers (GCs) following vaccination, suggesting that affinity maturation of these lineages against vaccine antigens can occur. However, it remains to be determined whether seasonal influenza vaccination stimulates additional evolution of vaccine-specific lineages, and previous work has found no significant increase in somatic hypermutation among influenza-binding lineages sampled from the blood following seasonal vaccination in humans. Here, we investigate this issue using a phylogenetic test of measurable immunoglobulin sequence evolution. We first validate this test through simulations and survey measurable evolution across multiple conditions. We find significant heterogeneity in measurable B cell evolution across conditions, with enrichment in primary response conditions such as HIV infection and early childhood development. We then show that measurable evolution following influenza vaccination is highly compartmentalized: while lineages in the blood are rarely measurably evolving following influenza vaccination, lineages containing GC B cells are frequently measurably evolving. Many of these lineages appear to derive from memory B cells. We conclude from these findings that seasonal influenza virus vaccination can stimulate additional evolution of responding B cell lineages, and imply that the poor efficacy of seasonal influenza vaccination is not due to a complete inhibition of vaccine-specific B cell evolution.

## Editor's evaluation

The manuscript by Hoehn et al., introduces a novel approach to measure evolution in B cell responses, and apply it to a wide variety of data sets. The work provides significant new insight into which stimuli induce effective immune responses, and which has the potential to improve vaccine design. This will be of interest to those interested in B cell responses, especially in the case of vaccinations that induce poor immune responses.

## Introduction

Measurably evolving populations are systems that undergo evolution rapidly enough for significant genetic differences to be detected in longitudinally sampled timepoints (*Drummond et al., 2003*). While this concept is frequently applied to viruses such as HIV (*Rambaut et al., 2004*) and SARS-CoV-2 (e.g., *du Plessis et al., 2021*), B cells experience similarly rapid evolution during affinity maturation. B cell affinity maturation is critical for developing high-affinity antibodies in response to infection and vaccination (*Shlomchik and Weisel, 2012*; *Victora and Nussenzweig, 2012*). During affinity maturation, somatic hypermutation (SHM) introduces mutations into the B cell receptor (BCR) loci at a rate orders of magnitude higher than the background rate of somatic mutations (*McKean et al., 1984*; *Murphy et al., 2008*). These modified BCRs are selected based on their binding affinity, and the process repeats cyclically within germinal centers (GCs *Teng and Papavasiliou, 2007*; *Victora and Nussenzweig, 2012*). Infection or vaccination can also stimulate pre-existing memory B cells that rapidly differentiate into antibody secreting plasmablasts or possibly re-enter GCs to undergo additional affinity maturation (*Ellebedy, 2018*; *Mesin et al., 2020*). A lack of vaccine-specific affinity maturation is thought to underlie the poor efficacy of seasonal influenza virus vaccination (*Arevalo et al., 2020*; *Ellebedy, 2018*). While recent work has shown that antigen-specific B cell lineages can be recruited into GCs following influenza vaccination (*Turner et al., 2020*), other work has been unable to detect significant increases in SHM frequency among circulating influenza-binding antibody lineages following vaccination (*Ellebedy et al., 2016*).

Whether seasonal influenza vaccination stimulates an increase in SHM frequency can be answered by determining whether influenza-binding B cell lineages found in GCs are measurably evolving following vaccination. This is distinct from simply quantifying SHM frequency. While influenza vaccination stimulates memory B cell lineages with high SHM frequency (*Laserson et al., 2014*; *Wrammert et al., 2008*), these lineages are only measurably evolving if their level of SHM detectably increases during the sampling interval surrounding vaccination. In this study, we show how a phylogenetic test of measurable evolution can be a powerful tool to detect increasing SHM frequency in longitudinally sampled BCR sequence datasets (*Duchêne et al., 2015*; *Murray et al., 2016*). We validate this approach through simulations and a survey of measurable evolution in B cell repertoires across a wide range of infections and vaccinations. We document significant heterogeneity among conditions, with some like HIV infection and primary hepatitis B vaccination enriched for measurably evolving lineages in the blood. We further show that while most circulating lineages following influenza virus vaccination are not measurably evolving, a subset of memory B cell lineages re-enter GCs and increase in SHM frequency.

## Results

### Detecting measurable evolution in longitudinally sampled BCR repertoires

We develop a framework to test for measurable evolution in B cells based on longitudinally sampled sequence data from the BCR variable region. After preprocessing the sequencing data, we first identify clonal lineages – B cells that descend from a common V(D)J rearrangement – using clustering based on nucleotide sequence similarity, which we have previously shown detects clonal relationships with high confidence (*Gupta et al., 2017*; *Zhou and Kleinstein, 2019*). The pattern of shared SHM among BCR sequences within a lineage is then used to build a B cell lineage tree, which represents a lineage's history of SHM. Branch lengths within these trees represent SHM per site. The divergence of each tip is the sum of branch lengths leading back to the lineage's most recent common ancestor. In evolving lineages, sequences sampled at later timepoints are expected to have higher divergence than those from earlier timepoints (*Figure 1A*). To estimate the rate of evolution over time, we calculate the slope of the regression line between timepoint (weeks) and divergence (SHM/site) for each tip (*Figure 1B, E*; *Rambaut et al., 2016*). Because tips are not independent, standard linear regression p values are improper. We instead quantify significance using a modified phylogenetic date randomization test (*Duchêne et al., 2015*; *Murray et al., 2016*). This tests whether the Pearson's correlation between divergence and time is significantly greater than that observed in the same tree with timepoints randomized among tips (*Figure 1C, F*). To account for population structure and sequencing error, we permute timepoints among single-timepoint monophyletic clusters of tips rather than

**eLife digest** When the immune system encounters a disease-causing pathogen, it releases antibodies that can bind to specific regions of the bacterium or virus and help to clear the infection. These proteins are generated by B cells which, upon detecting the pathogen, can begin to mutate and alter the structure of the antibody they produce: the better the antibody is at binding to the pathogen, the more likely the B cell is to survive. This process of evolution produces B cells that make more effective antibodies. After the infection, some of these cells become 'memory B cells' which can be stimulated in to action when the pathogen invades again.

Many vaccines also depend on this process to trigger the production of memory B cells that can fight off a specific disease-causing agent. However, it is unclear to what extent memory B cells that already exist are able to continue to evolve and modify their antibodies. This is particularly important for the flu vaccine, as the virus that causes influenza rapidly mutates. To provide high levels of protection, the memory B cells formed following the vaccine may therefore need to evolve to make different antibodies that recognize mutated forms of the virus.

It is thought that the low effectiveness of the flu vaccine is partially because the response it triggers does not stimulate additional evolution of memory B cells. To test this theory, Hoehn et al. developed a computational method that can detect the evolution of B cells over time. The tool was applied to samples collected from the blood and lymph nodes (organ where immune cells reside) of people who recently received the flu vaccine. The results were then compared to B cells taken from people after different infections, vaccinations, and other conditions.

Hoehn et al. found the degree to which B cells evolve varies significantly between conditions. For example, B cells produced during chronic HIV infections frequently evolved over time, while such evolution was rarely observed during the autoimmune disease myasthenia gravis. The analysis also showed that memory B cells produced by the flu vaccine were able to evolve if recruited to the lymph nodes, but this was rarely detected in B cells in the blood.

These findings suggest the low efficacy of the flu vaccine is not due to a complete lack of B cell evolution, but likely due to other factors. For instance, it is possible the evolutionary process it stimulates is not as robust as in other conditions, or is less likely to produce long-lived B cells that release antibodies. More research is needed to explore these ideas and could lead to the development of more effective flu vaccines.

individual tips (*Figure 1—figure supplements 1 and 2*; *Duchêne et al., 2015*; *Murray et al., 2016*). Further, it is possible that the combined effects of PCR and sequencing error will generate tree structures with multiple spurious tips radiating from a single node. This could increase the error rate of the date randomization test. Because trees are strictly binary, this would produce clusters of zero-length branches (soft polytomies) that could increase the error rate. To limit potential effects of this source of error, we resolve polytomies into the fewest number of single-timepoint monophyletic clades possible (*Figure 1—figure supplements 1 and 2*). We refer to lineages with a date randomization test p < 0.05 as 'measurably evolving'. To limit our analyses to lineages with adequate statistical power, we include only lineages with ≥15 total sequences sampled over at least 3 weeks, and have a minimum possible p value <0.05 based on the number of distinct permutations. Because we use a p value cutoff of 0.05, we expect a false positive rate of approximately 5% if no measurable evolution is occurring. We therefore refer to datasets with >5% measurably evolving lineages as 'enriched' for measurable evolution. This test is implemented within the Immcantation.org framework in the R package *dowser* (*Hoehn et al., 2020*).

To determine the necessary sampling interval to detect B cell evolution, we benchmarked the date randomization test using affinity maturation simulations performed with the package *bcr-phylo* (*Davidsen and Matsen, 2018*; *Ralph and Matsen, 2020*). This simulates alternating GC cycles of B cell proliferation, SHM, and selection based on amino acid similarity to a target sequence. Within these simulations, each lineage was first sampled after 10 simulated GC cycles, and then sampled a second time after a variable number of additional cycles. Using this framework in which all lineages are evolving, the date randomization test detected measurable evolution in 47% of lineages after 10 additional GC cycles, and 77% after 15 additional cycles (*Figure 1G*). Given a GC cycle time of

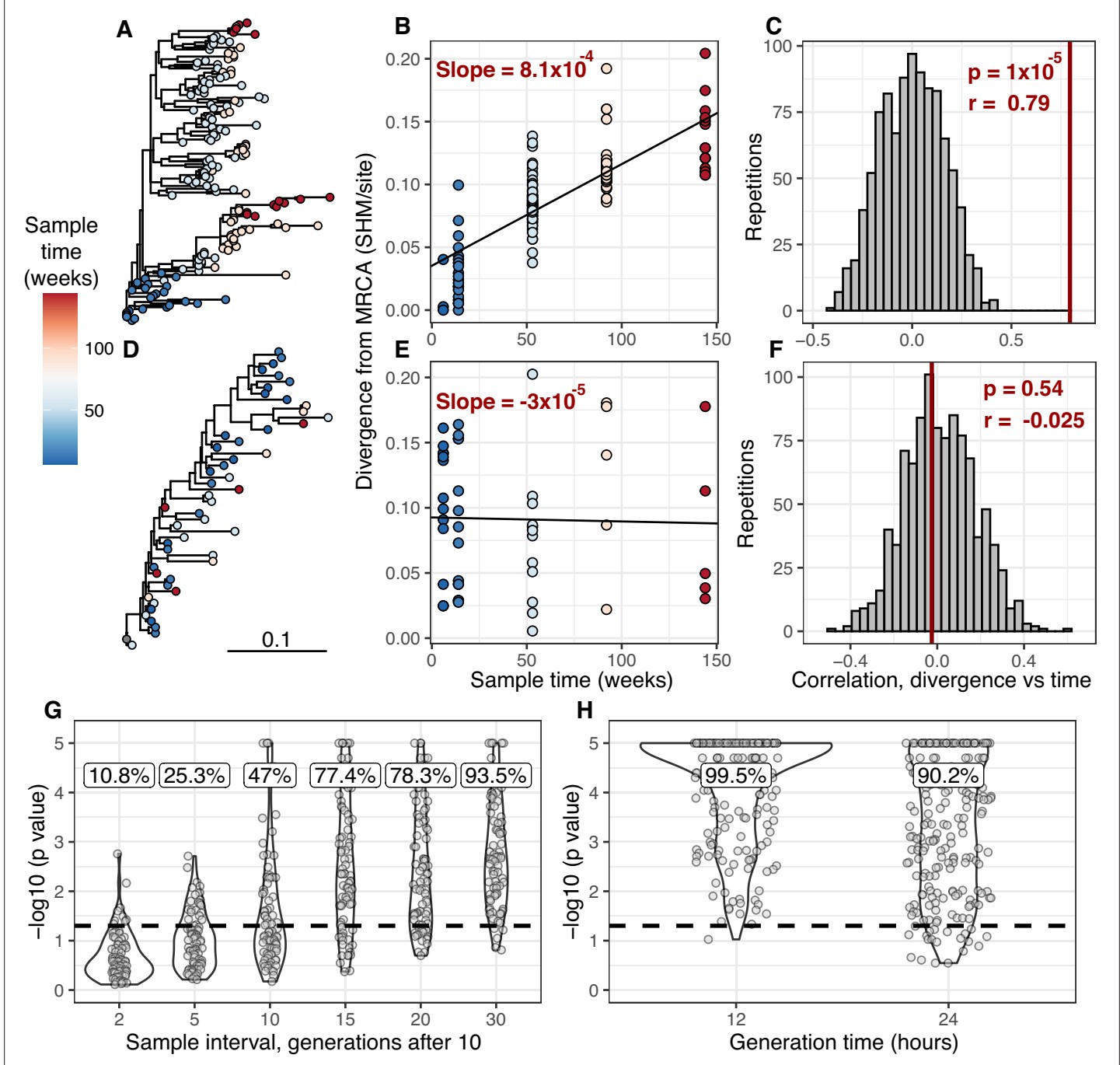

**Figure 1.** Detecting measurable evolution in B cell lineages. (**A**) Example B cell lineage tree from *Liao et al., 2013* showing increasing divergence with sample time. Branch lengths show somatic hypermutation (SHM)/site according to scale bar in (**D**). (**B**) Rate of SHM accumulation over time estimated using a regression of divergence vs time in tree (**A**). (**C**) Significance of the relationship between divergence and time estimated using a date randomization test comparing the Pearson's correlation (*r*) between divergence and time in tree (**A**). (**D–F**) Same plots as (**A–C**) but on a tree that is not measurably evolving. (**G**) Simulation-based power analysis shows the permutation test has high power over an interval of at least 10–30 GC cycles (generations). Lineages were sampled once at generation 10, and a second time after the specified number of additional generations have elapsed. Percentage of lineages with p < 0.05 are listed above, rounded to three significant digits. The dotted line corresponds to p = 0.05. (**H**) Simulation-based analysis reproducing the sampling of *Laserson et al., 2014* shows the test has high power even at slow (24 hr) GC cycle times.

The online version of this article includes the following figure supplement(s) for figure 1:

**Figure supplement 1.** Clustered date randomization and resolution of polytomies.

**Figure supplement 2.** Comparison of date randomization strategies.

*Figure 1 continued on next page*

*Figure 1 continued*

**Figure supplement 3.** Simulation-based power analysis.

**Figure supplement 4.** Simulation-based power analysis recreating experimental sampling design.

**Figure supplement 5.** Mean divergence of lineages from simulations in *Figure 1—figure supplement 3* under neutral evolution (selection = 0) or strong selection (selection = 1).

6–24 hr, 15 cycles corresponds to 4–15 days, within the timeframe of many longitudinal B cell repertoire studies (*Ellebedy et al., 2016*; *Laserson et al., 2014*). Interestingly, the date randomization test had higher power to detect measurable evolution in simulations of neutral evolution than those that included selection (*Figure 1—figure supplements 3–4*). This is likely because selection can reduce the rate of divergence within lineages compared to neutral evolution (*Figure 1—figure supplement 5*). To quantify the false positive rate, we repeated these calculations on the same simulations but with randomized sample time associations. Here, the date randomization test found measurable evolution in <4% in each case, indicating a low false positive rate (*Figure 1—figure supplements 3–4*). These analyses demonstrate that the date randomization test has sufficient sensitivity and specificity to detect ongoing B cell evolution from longitudinally sampled BCR data.

## Primary immune responses are enriched for measurably evolving lineages

To further validate our approach, we tested for measurable evolution in cases of known or suspected affinity maturation in humans. We hypothesized that primary immune responses would be enriched for measurably evolving lineages. To test this, we used publicly available data primarily from the Observed Antibody Space (OAS) database (*Kovaltsuk et al., 2018*) to survey measurable evolution in BCR datasets from 99 human subjects in 21 studies spanning 10 conditions including HIV infection, Ebola virus infection, and healthy controls (*Table 1*). We observed considerable heterogeneity in measurable evolution among conditions. Confirming our hypothesis, we observed an enrichment of measurably evolving lineages (>5% of tested lineages) in primary immune responses including HIV infection, meningococcus vaccination, primary but not secondary hepatitis B vaccination, and early childhood development (*Table 1*, *Figure 2A*, and *Figure 2—figure supplement 1*).

Chronic HIV infection stimulates ongoing affinity maturation as B cells evolve to contain viral escape mutants (*Liao et al., 2013*; *Wendel et al., 2020*). Consistent with this arms race, HIV infection was more enriched for measurably evolving lineages than other conditions surveyed, with each study having between 5.9% and 53% of lineages measurably evolving (*Figure 2A*). Lineages from subjects with broadly neutralizing anti-HIV lineages sampled over multiple years (*Doria-Rose et al., 2014*; *Landais et al., 2017*; *Liao et al., 2013*; *Wu et al., 2015*) were particularly enriched (26–53% measurably evolving). Importantly, the HIV studies included were sampled over longer time periods than studies of other conditions (mean = 225 vs 45 weeks, *Table 1*). To determine whether these results were simply due to longer sampling intervals, we repeated our analysis of subjects with HIV using only samples within the first 60 weeks of the study. These truncated datasets were still highly enriched for measurably evolving lineages (6.9–64%) compared to other non-HIV datasets with similar sampling intervals (0–7.2%, *Figure 2A*). This indicates that the observed high frequency of measurably evolving lineages is not simply due to long sampling intervals.

Other primary immune responses were also enriched for measurably evolving lineages (*Table 1*, *Figure 2A*). B cell lineages from healthy children sampled during the first 3 years of life were enriched for measurable evolution (14%), possibly reflecting continual exposure to novel antigens (*Nielsen et al., 2019*). We also observed an enrichment of measurably evolving lineages following primary meningococcus vaccination (10%; *Galson et al., 2015a*) and primary but not secondary hepatitis B vaccination (7.2% vs 2.9%, respectively; *Galson et al., 2016*; *Galson et al., 2015b*). Primary hepatitis B vaccinees were sampled over a longer time period than secondary vaccines, so this difference may also be due to different sampling intervals (*Figure 2—figure supplement 2*). Further, allergen-specific immunotherapy, which stimulates tolerance of allergy-causing antigens through exposure, was also enriched for measurable evolution (6.5%; *Levin et al., 2016*). Interestingly, Ebola virus infection showed a borderline (5%) percentage of measurably evolving lineages (*Table 1*) despite likely being a

**Table 1.** Summary of datasets.

*N* shows number of subjects with at least one powered lineage. *Mean range* shows mean total sampling interval across subjects. *Powered lineages* shows the number of lineages that: (1) contained at least 15 sequences, (2) were sampled over at least 3 weeks, and (3) had a minimum possible p value <0.05. The rightmost column shows the percentage of these lineages with p < 0.05, rounded to two significant digits. Studies with at least 5% positive lineages are shown in bold. *Turner et al., 2020* in this table and *Figure 2* included only blood samples. Data from studies marked with an asterisk (*) were obtained from Observed Antibody Space (*Kovaltsuk et al., 2018*).

| Study | Condition | N | Mean range (weeks) | Mean sample count | Multi-timepoint lineages | Powered lineages | % lineages p < 0.05 |
|---|---|---|---|---|---|---|---|
| **Levin et al., 2016*** | Allergy + SIT | 9 | 52 | 2.7 | 42 | 31 | 6.5 |
| **Davis et al., 2019*** | Ebola virus | 4 | 36 | 3.6 | 1,549 | 877 | 5 |
| **Wang et al., 2014** | Healthy adults | 7 | 52 | 2 | 18 | 10 | 0 |
| **Nielsen et al., 2019** | Healthy children | 20 | 69 | 2.7 | 262 | 71 | 14 |
| **Galson et al., 2015b*** | Hep. B vaccine (boost) | 9 | 4 | 4.8 | 4,923 | 3,422 | 2.9 |
| **Galson et al., 2016*** | Hep. B vaccine (primary) | 9 | 23 | 6.9 | 4,426 | 2,529 | 7.2 |
| **Doria-Rose et al., 2014*** | | 1 | 190 | 8 | 65 | 48 | 44 |
| **Huang et al., 2016*** | | 1 | 120 | 12 | 388 | 221 | 5.9 |
| **Johnson et al., 2018*** | | 1 | 170 | 5 | 561 | 330 | 23 |
| **Landais et al., 2017*** | | 1 | 160 | 7 | 1,084 | 743 | 48 |
| **Liao et al., 2013*** | | 1 | 140 | 5 | 205 | 151 | 53 |
| **Schanz et al., 2014*** | | 1 | 120 | 3 | 147 | 54 | 11 |
| **Setliff et al., 2018*** | | 6 | 170 | 3 | 787 | 173 | 9.8 |
| **Wu et al., 2011, Wu et al., 2015*** | HIV | 1 | 730 | 7 | 393 | 305 | 26 |
| **Ellebedy et al., 2016*** | | 8 | 13 | 5 | 1966 | 1,479 | 5.2 |
| **Laserson et al., 2014*** | | 3 | 4 | 9 | 1,182 | 639 | 4.9 |
| **Turner et al., 2020** | Influenza vaccine | 1 | 8.6 | 5 | 168 | 104 | 2.9 |
| **Galson et al., 2015a*** | Meningococcus vaccine | 7 | 4 | 3 | 483 | 80 | 10 |

*Table 1 continued on next page*

*Table 1 continued*

| Study | Condition | N | Mean range (weeks) | Mean sample count | Multi-timepoint lineages | Powered lineages | % lineages p < 0.05 |
|---|---|---|---|---|---|---|---|
| *Jiang et al., 2020a* | | 3 | 260 | 3.6 | 110 | 62 | 3.2 |
| *Jiang et al., 2020b* | Myasthenia gravis | 1 | 52 | 2 | 46 | 33 | 3 |
| *Tsioris et al., 2015* | West Nile virus | 6 | 5.2 | 2 | 151 | 65 | 1.5 |

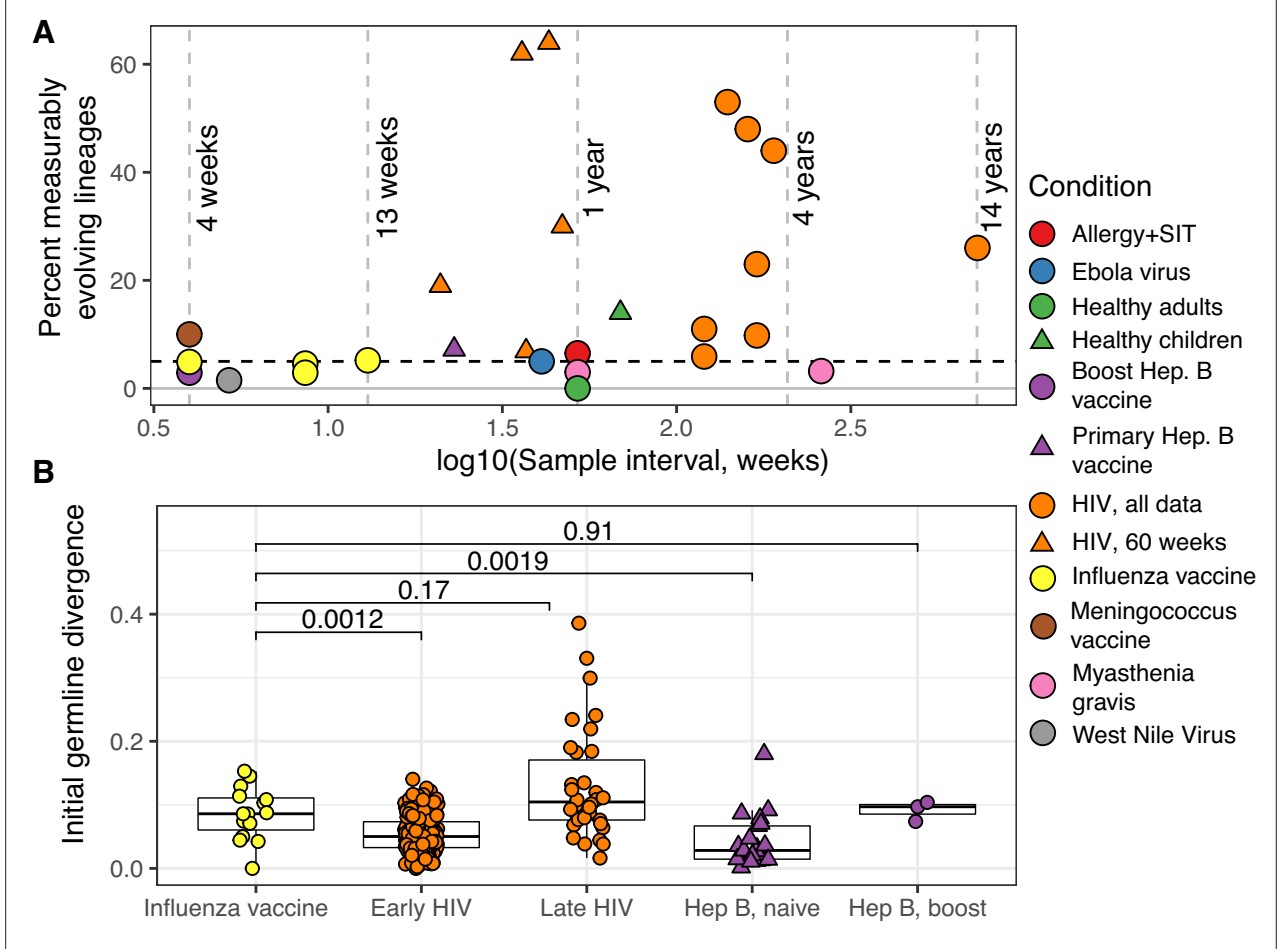

**Figure 2.** Measurable evolution in B cell lineages across time and conditions. (**A**) Percentage of lineages that are measurably evolving within each study (Table 1, *Figure 1C*). The dotted line indicates 5%, the percent expected under the null hypothesis that there is no measurable evolution occurring in a given dataset. Orange triangles indicate HIV datasets truncated to only include data within the first 60-week sampling interval. Note that three HIV studies were not truncated because they contained <2 sample timepoints within the first 60 weeks of sampling (*Huang et al., 2016*; *Schanz et al., 2014*; *Wu et al., 2015*). (**B**) Mean initial germline divergence (sum of branch lengths) from germline to sequences from each adjusted measurably evolving lineage's first timepoint. Note that 'Early/Late' HIV in (**B**) separates studies by time since initial infection, while 'HIV, first 60 weeks' in (**A**) includes only samples taken over the first 60 weeks of sampling. Each point is a measurably evolving lineage with a Benjamini–Hochberg adjusted p value <0.1. Wilcoxon tests were used to compare divergence levels among datasets.

The online version of this article includes the following figure supplement(s) for figure 2:

**Figure supplement 1.** Date randomization p value histograms from blood-derived lineages across all studies.

**Figure supplement 2.** Comparison of measurably evolving lineages among hepatitis B vaccine schedules.

**Figure supplement 3.** Enrichment of antigen-binding monoclonal antibody (mAb) sequences across studies.

**Figure supplement 4.** Initial germline divergence with alternate p value thresholds.

primary infection. Overall, however, these results confirm that the date randomization test can detect ongoing SHM in empirical datasets where it is expected to be occurring.

We next investigated whether measurable evolution was associated with antigen-binding lineages. While antigen-binding information was not available for most B cell lineages surveyed, some studies included experimentally validated monoclonal antibody sequences (mAbs). Lineages containing these sequences thus contain B cells that bind to the antigen under study. Experimentally validated mAbs were included from six studies: four in HIV (*Doria-Rose et al., 2014*; *Landais et al., 2017*; *Liao et al., 2013*; *Wu et al., 2015*), one in Ebola virus infection (*Davis et al., 2019*), and one in influenza vaccine response (*Turner et al., 2020*). We found that across these studies measurably evolving lineages were more likely to contain mAbs than nonmeasurably evolving lineages (p = 0.031, Wilcoxon test,

*Figure 2—figure supplement 3*). This is consistent with the hypothesis that measurably evolving lineages are actively responding to antigens relevant to the condition being studied.

## Measurably evolving lineages are rare in peripheral blood following influenza vaccination

Seasonal influenza vaccination is believed to trigger a memory B cell response in adults. If memory B cells rarely re-enter GCs to undergo additional affinity maturation (*Mesin et al., 2020*), and there is little evolution of naive B cell lineages, we expect little measurable evolution in the blood following vaccination. To test this, we applied the date randomization test to three longitudinally sampled adult influenza vaccine datasets. The first comprised three adults sampled seven times between 1 hr and 28 days postvaccination (*Gupta et al., 2017*; *Laserson et al., 2014*); the second contained eight adults sampled five times between 0 and 90 days postvaccination (*Ellebedy et al., 2016*) the third used blood samples from a single individual sampled five times between 0 and 60 days postvaccination (*Turner et al., 2020*). Across subjects in each study, between only 2.9% and 5.2% of lineages were measurably evolving (*Table 1*). These values are approximately as expected under the null hypothesis of no measurable evolution, and histograms of p values from these datasets are roughly uniform, suggesting the measurably evolving lineages identified are mostly false positives from multiple testing (*Figure 2—figure supplement 1*). Distributions of p values for all datasets are also available in *Figure 2—figure supplement 1*. To verify the 4- to 13-week sampling range of these studies was sufficient to detect measurable evolution, we performed simulation analyses replicating the sampling strategy of the influenza dataset with the shortest sampling range (*Figure 1—figure supplement 4*; *Laserson et al., 2014*). These simulations show this timescale was sufficiently long to detect ongoing affinity maturation with high sensitivity (>90%, *Figure 1H*). Overall, these results indicate B cell lineages present in blood infrequently undergo additional evolution within 13 weeks following influenza vaccination, consistent with a primarily GC-independent memory B cell response and/or rarity of antigen-specific lineages in the peripheral blood (*Wrammert et al., 2008*).

## Measurably evolving lineages following influenza vaccination include memory B cell origin

While measurably evolving lineages do not occur at high frequency in the blood following influenza vaccination, we checked if any could be identified after adjustment for multiple testing. To adjust for multiple hypothesis tests, we pooled lineages across all studies and adjusted their p values using the Benjamini–Hochberg procedure (*Benjamini and Hochberg, 1995*). We identified 15 lineages in influenza datasets, and 354 lineages in other conditions, with adjusted date randomization p values < 0.1. We investigated if these 'adjusted' measurably evolving lineages were derived from naive or pre-existing memory B cells. Because memory B cell lineages have already undergone affinity maturation, they are expected to have higher initial SHM levels compared to naive B cell lineages. To test this, we compared germline sequence divergence in adjusted measurably evolving lineages from influenza vaccination to other conditions. Consistent with memory B cell reactivation, lineages from influenza vaccination had significantly higher initial divergence (median = 8.6%) than those from primary responses such as early HIV infection (median = 5%, p = 0.0012) and primary hepatitis B vaccination (median = 2.8%, p = 0.0019) (*Figure 2B*). Further, these influenza lineages had initial divergence levels similar to lineages from subjects with HIV first sampled >5 years after infection (*Huang et al., 2016*; *Wu et al., 2015*), and hepatitis B booster vaccination subjects (*Figure 2B*; *Galson et al., 2015b*). Ebola virus infection, meningococcus vaccination, and early childhood development had median initial divergence levels of 0.4%, 6.6%, and 2.0%, respectively, but contained less than three adjusted measurably evolving lineages each. To understand the effect of multiple hypothesis correction on these results, we repeated the comparisons in *Figure 2B* using all measurably evolving lineages (unadjusted p < 0.05) from the same datasets. Considering this larger set of lineages, initial divergence of lineages from influenza vaccination studies was significantly higher than those in all other conditions except late HIV infection (*Figure 2—figure supplement 4*). The same pattern from *Figure 2B* was also found when repeating these comparisons with a more strict cutoff (adjusted p < 0.05, *Figure 2—figure supplement 4*). Overall, these results are consistent with measurably evolving lineages from influenza vaccination arising mainly from pre-existing memory B cells.

## Measurably evolving lineages show signs of purifying selection

We next investigated the type and degree of selection operating on measurably evolving B cell lineages. One way to detect natural selection in DNA sequences is to estimate the ratio of nonsynonymous (amino acid replacement) to synonymous (silent) mutation rates. This ratio is often called $\omega$ (*Nielsen and Yang, 1998*). Neutral evolution, where amino acid replacements are not selected for or against, should result in $\omega = 1$. Purifying selection, where amino acid replacements are disfavored, should result in $\omega < 1$. Diversifying selection, where amino acid replacements are favored, should result in $\omega > 1$. In B cell lineages, $\omega$ is often estimated separately for complementarity-determining regions (CDRs) involved in antigen binding, and framework regions (FWRs), which are more structural. Further, it is important to estimate $\omega$ or similar metrics using models that account for intrinsic hot- and cold-spot biases of SHM (*Hoehn et al., 2017*; *Uduman et al., 2011*; *Yaari et al., 2012*). To understand what kind of selection operated on measurably evolving lineages, we estimated separate $\omega$ values for CDR and FWR regions ($\omega_{CDR}$ and $\omega_{FWR}$) of the adjusted measurably evolving lineages (*Figure 2B*) using the HLP19 model in IgPhyML (*Hoehn et al., 2019*). Model parameters were shared among lineages within the same subject, and only subjects with at least two adjusted measurably evolving lineage were included to reduce noise. Across all conditions surveyed, we found evidence of purifying selection operating on adjusted measurably evolving lineages (mean $\omega_{CDR} = 0.58$, mean $\omega_{FWR} = 0.48$, *Table 2*). We estimated the significance of these results using a phylogenetic likelihood ratio test (*Huelsenbeck and Rannala, 1997*). We found that $\omega_{CDR}$ was significantly <1 in 10/13 subjects (significantly >1 in none) and $\omega_{FWR}$ was significantly <1 in 13/13 subjects (*Table 2*). This signal of purifying selection was particularly strong in both early and late HIV. Influenza vaccination showed higher $\omega$ values, comparable to primary hepatitis B vaccination.

## Influenza-binding lineages associated with GCs are measurably evolving

While we found little measurable evolution in the blood following seasonal influenza vaccination, influenza vaccination has been shown to stimulate both naive and memory B cells to enter GCs (*Turner et al., 2020*). This raises the possibility that additional affinity maturation could be occurring in GCs, but its products are not enriched in the blood. Data from *Turner et al., 2020* provided both blood samples and fine-needle aspirations of lymph nodes (including GCs) from the same subject. By combining these samples, we identified 53 powered B cell lineages containing at least one GC B cell following influenza vaccination, and 100 powered lineages that contained none. We refer to lineages containing one or more GC B cells as 'GC-associated'. To determine whether GC-associated lineages were undergoing additional SHM, we tested whether they were enriched for measurable evolution. We found that 7.5% of lineages containing sequences from GC B cells were measurably evolving, compared to only 3.0% of lineages with no identified GC sequences. This signal of measurable evolution increased with the fraction of GC sequences. For instance, while 10% of lineages containing ≥10% GC sequences were measurably evolving, 38% (3/8) of those with ≥25% GC sequences were measurably evolving (*Figure 3A*). Lineages with higher proportions of GC sequences also had a higher correlation between divergence and time (linear regression slope = 1.1, p = $8.9 \times 10^{-13}$, *Figure 3—figure supplement 1*). We further estimated the significance of this positive relationship by bootstrapping our data using 10,000 resampling repetitions with replacement. We found that in all 10,000 resampling repetitions, the slope of the linear regression between GC sequence proportion and the correlation between divergence and time was positive, with 95% of repetitions having a slope between 0.81 and 1.3 (*Figure 3—figure supplement 1*). Measurably evolving lineages in this dataset did not contain significantly more sequences than other lineages, indicating these results were not significantly confounded by lineage size (*Figure 3—figure supplement 2*). Finally, the measurably evolving lineages with the highest proportion of GC sequences contained mAbs that bound to vaccine antigens (*Figure 3B, C*). These lineages show signs of origin from memory B cells, such as clonal relatedness to blood plasmablasts sampled 5 days postvaccination, and high mean germline divergence at their first sampled timepoint (6.3%, 7.2%, *Figure 3B, C*, respectively). To test whether GC-associated lineages accumulated new amino acid replacement mutations rather than just silent mutations, we repeated the date randomization test but calculated the divergence of each tip as the number of amino acid differences between that tip's sequence and the unmutated germline ancestor. This amino acid-based correlation analysis also showed a strong positive relationship between the

**Table 2.** Analysis of selection on adjusted measurably evolving lineages.

Repertoire-wide estimates of $\omega$ for CDRs ($\omega_{CDR}$) and FWRs ($\omega_{FWR}$) for adjusted measurably evolving lineages within different subjects are shown. $L$ indicates the maximum log-likelihood obtained when both $\omega_{CDR}$ and $\omega_{FWR}$ were estimated by maximum likelihood. $L_{CDR=1}$ indicates the maximum log-likelihood obtained when $\omega_{FWR}$ was estimated by maximum likelihood but $\omega_{CDR}$ was fixed at 1. $L_{FWR=1}$ indicates the maximum log-likelihood obtained when $\omega_{CDR}$ was estimated by maximum likelihood but $\omega_{FWR}$ was fixed at 1. The likelihood ratio statistic (LRS) was calculated as either $2\times(L - L_{CDR=1})$ for CDRs or $2\times(L - L_{FWR=1})$ for FWRs, and p values were calculated using a likelihood ratio test with one degree of freedom (see Methods). $L$ values were rounded to two decimal places, LRS values are reported to three significant digits. Significant p values are in bold. p values below the numerical limit for double values are reported as <2E−16.

| | Study | Subject | N | Region | $\omega$ | $L$ | $L_{CDR=1}$ | $L_{FWR=1}$ | LRS | p |
|---|---|---|---|---|---|---|---|---|---|---|
| Influenza vaccine | Ellebedy et al., 2016 | Donor-4 | 4 | CDR | 0.624 | −5753.04 | −5759.65 | | 13.2 | **2.80E−04** |
| | | | | FWR | 0.503 | −5753.04 | | −5783.9 | 61.7 | **4.00E−15** |
| | | Donor-5 | 5 | CDR | 0.979 | −5074.01 | −5074.02 | | 0.0212 | 8.84E−01 |
| | | | | FWR | 0.584 | −5074.01 | | −5090.55 | 33.1 | **8.75E−09** |
| | Laserson et al., 2014 | FV | 4 | CDR | 0.583 | −9162.11 | −9173.39 | | 22.6 | **1.99E−06** |
| | | | | FWR | 0.508 | −9162.11 | | −9217.38 | 111 | **<2E−16** |
| Early HIV | Doria-Rose et al., 2014 | CAP256 | 9 | CDR | 0.424 | −18,976.96 | −19,045.96 | | 138 | **<2E−16** |
| | | | | FWR | 0.451 | −18,976.96 | | −19,151.29 | 349 | **<2E−16** |
| | Johnson et al., 2018 | CAP256 | 26 | CDR | 0.423 | −46,720.28 | −46,858.67 | | 277 | **<2E−16** |
| | | | | FWR | 0.408 | −46,720.28 | | −47,243.96 | 1,050 | **<2E−16** |
| | Landais et al., 2017 | PC064 | 188 | CDR | 0.39 | −416,489.06 | −418,035.9 | | 3,090 | **<2E−16** |
| | | | | FWR | 0.386 | −416,489.06 | | −422,043.68 | 11,100 | **<2E−16** |
| | Liao et al., 2013 | CH505 | 51 | CDR | 0.527 | −102,131.66 | −102,307.3 | | 351 | **<2E−16** |
| | | | | FWR | 0.417 | −102,131.66 | | −103,240.86 | 2,220 | **<2E−16** |
| | Schanz et al., 2014 | ZA159 | 2 | CDR | 0.657 | −5573.47 | −5577.12 | | 7.29 | **6.93E−03** |
| | | | | FWR | 0.466 | −5573.47 | | −5617.91 | 88.9 | **<2E−16** |

*Table 2 continued on next page*

Table 2 continued

| | Study | Subject | N | Region | $\omega$ | L | $L_{CDR=1}$ | $L_{FWR=1}$ | LRS | p |
|---|---|---|---|---|---|---|---|---|---|---|
| | *Huang et al., 2016* | Donor-Z258 | 2 | CDR | 0.345 | −3162.12 | −3172.87 | | 21.5 | 3.54E−06 |
| | | | | FWR | 0.401 | −3162.12 | | −3196.5 | 68.7 | 1.11E−16 |
| Late HIV | *Wu et al., 2015* | Donor-45 | 32 | CDR | 0.451 | −63,527.81 | −63,687 | | 318 | <2E−16 |
| | | | | FWR | 0.378 | −63,527.81 | | −64,444.73 | 1,830 | <2E−16 |
| | | Subject-2277 | 6 | CDR | 0.885 | −5498.77 | −5499.15 | | 0.766 | 3.81E−01 |
| | | | | FWR | 0.668 | −5498.77 | | −5508.36 | 19.2 | 1.18E−05 |
| | | Subject-2752 | 2 | CDR | 0.533 | −1112.1 | −1113.82 | | 3.44 | 6.36E−02 |
| | | | | FWR | 0.461 | −1112.1 | | −1120.06 | 15.9 | 6.68E−05 |
| Hep B. vaccine, primary | *Galson et al., 2016* | Subject-2954 | 9 | CDR | 0.711 | −6023.13 | −6026.4 | | 6.56 | 1.04E−02 |
| | | | | FWR | 0.545 | −6023.13 | | −6048.08 | 49.9 | 1.62E−12 |

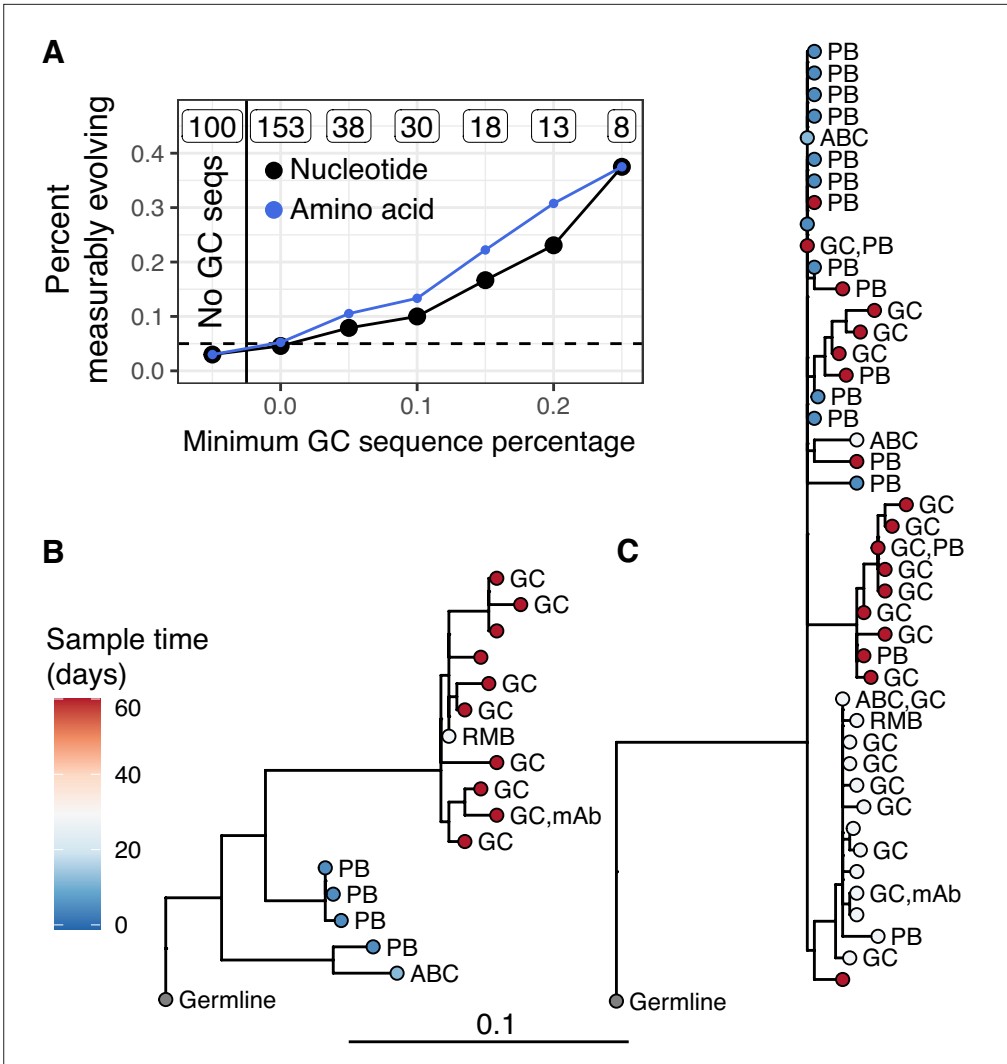

**Figure 3.** Germinal center (GC) association is positively related to measurable evolution following influenza vaccination. (**A**) Percent of lineages that are measurably evolving given a minimum percentage of GC sequences. The minimum (inclusive) percent of GC sequences within a clone is shown on the *x* axis. The origin shows the percentage of measurably evolving lineages across all lineages. The left-most point shows lineages without any GC sequences. The total number of lineages in each category are listed above each point. The dashed line shows 5%, the expected false positive rate under the null hypothesis. Results are shown for clustered date randomization tests using divergence values calculated either as the sum of nucleotide-based phylogenetic branch lengths (nucleotide), and the amino acid Hamming distance of each sequence to the germline (amino acid). (**B, C**) Lineage trees showing measurably evolving lineages with the highest proportion of GC sequences. Tips are labeled by cell type if available. ABC, activated B cell; GC, germinal center; PB, plasmablast; RMB, resting memory B; and unlabeled tips are from bulk PBMC sequencing. mAb = influenza-binding monoclonal antibody sequence (2018/2019 quadrivalent inactivated influenza virus vaccine). Branch lengths represent somatic hypermutation (SHM)/site, as shown by the shared scale bar.

The online version of this article includes the following figure supplement(s) for figure 3:

**Figure supplement 1.** Germinal center (GC) engagement is positively related to measurable evolution following influenza vaccination.

**Figure supplement 2.** Number of sequences per lineage in measurably evolving vs nonmeasurably evolving lineages.

**Figure supplement 3.** Percent of lineages that are measurably evolving given a minimum percentage of germinal center (GC) sequences.

proportion of lineages that were measurably evolving and the percentage of sequences derived from GC B cells (*Figure 3A*). This indicates that these GC-associated lineages accumulated new amino acid mutations as well as nucleotide mutations over the study interval. Overall, these analyses demonstrate that influenza-binding, GC-associated B cell lineages undergo additional, measurable evolution following vaccination.

A possible alternative explanation for measurable evolution following influenza vaccination is that SHM is not occurring over the sampled time interval, but that highly mutated B cells were preferentially recalled due to their higher binding affinity. Preferential recall of highly mutated B cells would likely result in a positive correlation between divergence and sample time. While difficult to directly test, we believe this explanation is unlikely to be the sole source of measurable evolution in our data. First, blood samples taken 5 days postvaccination represent the breadth of the pool of memory B cells. If measurable evolution was simply due to expansion of mutated memory B cells, we would expect divergence of later-sampled B cells to be within the range of day 5 plasmablasts. Instead, many later-sampled GC sequences are clearly more diverged than earlier-sampled sequences within the measurably evolving, influenza-binding lineages we observed (*Figure 3B, C*). Second, if measurable evolution were due simply to preferential expansion of more mutated B cells, we would expect to observe measurable evolution within influenza-binding lineages in both the blood and GC. This is not tested in *Figure 3A* because that analysis includes all lineages, not just those that bind to influenza. To adjust for this, we repeated the analysis in *Figure 3A* while only including lineages that contained influenza-binding mAbs. Even among influenza-binding lineages, we still observed an association between GC cells and measurable evolution. While 1/10 of influenza-binding lineages found only in the blood were measurably evolving, 2/5 lineages with >25% GC sequences were measurably evolving (*Figure 3—figure supplement 3*). Overall, while we cannot rule out preferential expansion of highly mutated memory B cells, these results are more easily interpretable as the result of ongoing SHM in GCs.

## Discussion

The extent to which seasonal influenza vaccination stimulates affinity maturation against vaccine antigens is unclear, and poor efficacy of seasonal influenza virus vaccination is often attributed to stimulation of pre-existing memory B cells interfering with novel responses to vaccine antigens (*Ellebedy, 2018*). While a prior study has shown that influenza-binding B cell lineages are found in GCs following seasonal influenza vaccination (*Turner et al., 2020*), other work has suggested that circulating influenza-binding B cell lineages do not accumulate additional SHM following vaccination (*Ellebedy et al., 2016*). To determine whether seasonal influenza vaccination stimulates additional evolution in B cell lineages, we developed and validated a framework to detect measurable evolution using longitudinally sampled BCR sequencing data. This phylogenetic test can be a powerful tool to detect ongoing B cell evolution using longitudinally sampled BCR datasets across a wide array of immunological conditions, including influenza virus vaccine responses. Our results confirm prior findings that there is little evidence of B cell evolution among lineages sampled in the peripheral blood following seasonal influenza vaccination (*Ellebedy et al., 2016*). However, we also show that seasonal influenza vaccination is capable of stimulating measurable evolution in influenza-binding, GC-associated B cell lineages.

To place our analyses of the influenza vaccination response in a broader context, we surveyed measurable evolution across a broad range of infections and vaccinations. Prior work has shown that chronic HIV infection induces long-term affinity maturation of broadly neutralizing antibody lineages in response to viral escape mutants (*Liao et al., 2013*; *Vieira et al., 2018*; *Wu et al., 2015*). Our results show that HIV infection is associated with an exceptionally strong signature of B cell evolution over time. This signature is not limited to single lineages. Rather, a substantial fraction of longitudinally sampled B cell lineages within the repertoires of subjects with HIV are measurably evolving, consistent with clonal competition among B cell lineages during HIV infection (*Nourmohammad et al., 2019*). Early childhood development during the first 3 years of life (*Nielsen et al., 2019*) showed the second-highest enrichment of measurably evolving lineages among surveyed conditions. This possibly reflects continual exposure to novel antigens during childhood. Further, primary vaccinations (meningococcus, primary hepatitis B)(*Galson et al., 2016*; *Galson et al., 2015a*) showed stronger signatures of measurable evolution than secondary vaccinations (adult seasonal influenza, hepatitis B booster) (*Ellebedy*

*et al., 2016*; *Galson et al., 2015b*; *Laserson et al., 2014*; *Turner et al., 2020*). Overall, our results are consistent with the hypothesis that GC responses are stronger in response to novel antigens.

In addition to detecting measurable evolution, we also characterized selection operating on measurably evolving B cell lineages. We found that measurably evolving lineages showed evidence of purifying selection ($\omega_{CDR} < 1$) (*Table 2*). Though perhaps counterintuitive, a strong signal of purifying selection is a straightforward prediction of evolution toward an adaptive peak (*Hoehn et al., 2019*). Similar evidence of purifying selection during affinity maturation has been observed in other studies including influenza vaccination, HIV infection, and healthy controls (*Cizmeci et al., 2021*; *Hoehn et al., 2019*; *Sheng et al., 2016*; *Yaari et al., 2015*). Importantly, $\omega$ estimates are an average across all codons within CDRs or FWRs. It is possible that positive selection operated on a small number of codon sites, but that this signal was outweighed by the larger number of sites under purifying selection. Codon-specific models may be useful in future analyses to identify these sites under positive selection (e.g., *Yang et al., 2000*). When interpreting these results, it is also important to note that parameters were estimated using all mutations represented by each lineage tree, including those that potentially occurred before the first sampled timepoint. In all, these results are consistent with typical forces of selection having operated on measurably evolving lineages.

There are several limitations to this study. Data from different studies were sampled according to different schedules and time intervals. Because the power to detect measurable evolution should increase over time (*Figure 1G*), this could confound comparisons among datasets. We note however that multiple influenza and HIV datasets were surveyed, and enrichment of measurable evolution within these conditions was not strongly related to the sampling range (*Figure 2A*). This suggests immunological condition, rather than sample range, was the primary determinant of observed differences. By including monoclonal antibody (mAb) sequences with experimentally validated binding in several datasets, we were able to show that measurably evolving lineages are more likely than nonmeasurably evolving lineages to contain mAb sequences. This is consistent with the idea that measurably evolving lineages are actively responding to antigen. These results should be interpreted cautiously, however. With the exception of *Turner et al., 2020* only a small number mAb sequences were found in the lineages analyzed (mean = 3.6 per study). Further, lineages containing GC sequences were preferentially selected for mAb generation in *Turner et al., 2020*, which may artificially increase the likelihood that measurably evolving lineages contain mAbs. While intriguing and biologically plausible, conclusively determining whether measurable evolution predicts antigen binding is beyond the scope of this study. Another limitation is that maximum parsimony was used to estimate lineage tree topologies and branch lengths. While more sophisticated methods are available for inferring B cell lineage trees (e.g., *Hoehn et al., 2019*), maximum parsimony often has competitive performance for topology estimation (*Davidsen and Matsen, 2018*) and is faster than more complex maximum likelihood models designed for B cell lineages. Computational efficiency was particularly important as our analyses required constructing more than 20,000 lineages trees spanning approximately 1,100,000 BCR sequences. Finally, our analysis of GC-associated B cell lineages was limited to data sampled from a single subject. Thus, while the results here demonstrate that influenza vaccination is capable of inducing measurable evolution, it remains unclear whether this is a general feature of influenza vaccination.

Our analyses of measurable evolution involved a series of hypothesis tests, and the definition for 'enrichment' of measurable evolution (>5% of lineages) was chosen based on the expected false positive rate under the null hypothesis. This enrichment measure was chosen to compare the relative frequency of measurably evolving lineages among datasets. A lack of enrichment does not indicate a complete lack of measurably evolving lineages. Conversely, slight enrichment of measurably evolving lineages (~5%) should not be interpreted as proof of ongoing affinity maturation in a set of lineages. The analysis in *Figure 2A* was not intended to test the null hypothesis that no lineages are measurably evolving in a particular dataset. Because multiple studies were surveyed (*Table 1*), it is possible our results contain false positives. Indeed, while some conditions such as HIV have a strong signal of measurable evolution across multiple studies, others such as naive hepatitis B vaccination and allergen-specific immunotherapy are just above the significance threshold (≤7.2% lineages with p < 0.05) and are each represented by a single study (*Figure 2A*, *Table 1*). These latter datasets should be interpreted cautiously and with the understanding that the vast majority of their lineages were not measurably evolving. To limit the influence of false positives, a multiple testing correction (false

discovery rate < 0.1) was performed in analyses investigating the properties of measurably evolving lineages (*Figure 2B*, *Table 2*). We note, however, that repeating our analyses of initial germline divergence using all measurably evolving lineages (unadjusted p < 0.05) or a more strict p value cutoff (adjusted p < 0.05) yielded similar results (*Figure 2—figure supplement 4*), indicating our results are robust to the thresholds used. Finally, we validated the specificity of the date randomization test in empirical data (*Figure 1—figure supplement 2*). While a significant decrease in divergence over time is biologically unlikely, false positives due to multiple testing or sequencing error should produce a similar number of lineages with a significant correlation in either direction. We quantified correlation in either direction using a two-tailed version of the clustered, resolved date randomization test with a critical value of 0.025 (see Methods). Under a null hypothesis of no ongoing evolution, 2.5% of lineages were expected to have a significant negative correlation between divergence and time (all false positives). However, we found such 'negatively evolving' lineages at a mean frequency of only 1.2% (median = 1.3%, range = 0–2.8%) across all datasets when using clustered permutations and resolved polytomies (*Figure 1—figure supplement 2*). This indicates the date randomization test we used is conservative, and that our chosen thresholds are likely more strict than necessary.

Beyond multiple testing, there are some biological scenarios that could plausibly give rise to a signal of measurable evolution without additional SHM occurring during the sampling interval. For instance, it is possible that all SHM within a lineage occurred before the first sampled timepoint in a study, but that more mutated, higher-affinity BCRs were preferentially stimulated and sampled in later timepoints. Such a scenario would likely result in a positive correlation between divergence and sample time. However, our results are easier to explain if measurable B cell evolution results at least in part from ongoing SHM. For instance, our analysis of measurably evolving lineages following influenza vaccination showed (1) many later-sampled GC B cells had higher divergence than any sampled day 5 plasmablast sequence in the same lineage and (2) continued association between GC cells (rather than the blood) and measurable evolution even among influenza-binding lineages. Nonetheless, it is still theoretically possible that more mutated B cells were not sampled at early timepoints, and were preferentially expanded in GCs compared to the peripheral blood. Thus, while our interpretation is that measurable B cell evolution most likely represents an ongoing SHM process, we cannot conclusively rule out biased selection of more mutated sequences that were generated before the sampling interval.

Affinity maturation is a rapid evolutionary process. It is perhaps surprising that, while we identified conditions enriched for measurably evolving lineages, most lineages in circulation were not measurably evolving (*Table 1*). One explanation is that our analyses did not use sufficiently long sampling intervals to detect affinity maturation, though we believe this is unlikely. Studies in mice have estimated that SHM occurs at ~$10^{-3}$ SHM/bp/division (*Kleinstein et al., 2003*; *McKean et al., 1984*), and that GC B cells cycle every 6–12 hr (*Allen et al., 2007*; *Hauser et al., 2007*; *Victora and Nussenzweig, 2012*). Simulations using conservative assumptions (strong selection, 24-hr cell cycle) and replicating the sample structure of our shortest-term influenza dataset (4 weeks), showed high power with >90% true positive rate (*Figure 1H*). Further, we found an enrichment of blood-derived measurably evolving lineages after only 4 weeks in one study (*Galson et al., 2015a*), and after 8 weeks corresponding to a known context of affinity maturation (GC entry, *Figure 3*). Overall, these results show that the sample times of our surveyed datasets should be sufficient to detect ongoing B cell evolution if it were occurring. However, it is possible that lineages may not remain in GCs continuously, which would slow the rate of evolution compared to our simulations. A more plausible explanation for the lack of measurable evolution is that most lineages in the blood are either nonspecific to the condition being studied, or derive from a GC-independent response (*Mesin et al., 2020*; *Takemori et al., 2014*; *Taylor et al., 2012*). It is also possible that lineages relevant to the condition being studied are inefficiently stimulated.

We find that seasonal influenza virus vaccination in young adults induces a GC reaction where maturation of vaccine-specific B cell lineages occurs, including those likely recruited from the pre-existing memory B cell compartment. These results imply that poor efficacy of seasonal influenza vaccination does not result from a complete lack of vaccine-induced B cell evolution. While we showed that B cells in these evolving lineages increased in amino acid replacement mutation frequency, it remains possible that this evolution is less able to select affinity-increasing mutations (*Hoehn et al., 2019*), that the overall number of evolving lineages is reduced, or that the products of this vaccine-induced

evolution are not efficiently translated into memory and long-lived plasma cells. These latter two explanations are consistent with the results of our survey of longitudinally sampled peripheral blood datasets, which found an enrichment of measurably evolving lineages in some primary immune response conditions, but not influenza vaccination. Future studies will be needed to fully test these hypotheses about the causes of poor efficacy of seasonal influenza vaccination.

## Materials and methods

### Study design

The goal of this study was to determine whether B cell lineages found in GCs following influenza vaccination evolved over a given sample interval. This necessitated describing and validating a test for measurable evolution from longitudinally sampled BCR sequencing data. Simulation-based power analyses determined that this date randomization test has sufficient sensitivity to detect evolving B cell populations over a sampling interval of approximately 2 weeks. To determine whether the date randomization test also worked on known examples of affinity maturation, all longitudinally sampled datasets hosted on OAS (as of 6/2020) were downloaded and tested. To cover as wide a variety of conditions as possible, these datasets were supplemented with processed, publicly available datasets from other prior studies. To ensure datasets were appropriately powered, datasets were only included if they contained at least 10 B cell lineages with at least 15 sequences sampled over 3 weeks and a minimum possible date randomization test p value <0.05. BCR data from blood and fine-needle aspirations following influenza vaccination were obtained from *Turner et al., 2020*.

### BCR sequence datasets and preprocessing

All longitudinally sampled BCR repertoire datasets were publicly available and obtained both from primary publications and through the OAS database (antibodymap.org, accessed 6/2020; *Kovaltsuk et al., 2018*). Both assembled nucleotide sequences and deduplicated amino acid sequences were obtained from OAS. To reduce the effect of sequencing error in OAS datasets, only nucleotide sequences corresponding to an amino acid sequence with a multiplicity of at least two were included. Datasets obtained from OAS are labeled in *Table 1*. Raw sequence data obtained from *Nielsen et al., 2019* were preprocessed with pRESTO v0.5.13 (*Vander Heiden et al., 2014*). Quality control was performed by first removing all sequences with a Phred quality score <20, length <300 bp, or any missing ('N') nucleotides. The 3′ and 5′ ends of each read were matched to forward and constant region primers with a maximum error rate of 0.1. The region adjacent to the constant region primer was exactly matched to subisotype-specific internal constant region sequences. Only sequences with the same isotype predicted by their constant region primer and internal constant region sequence were retained. Identical reads within the same isotype were collapsed and sequences observed only once were discarded. All other datasets used processed BCR sequence data provided by the authors of their respective publications. Data from *Wang et al., 2014* were processed in *Hoehn et al., 2019*. Data from *Jiang et al., 2020b* used only blood samples.

### BCR sequence processing, genotyping, and clonal clustering

Datasets were processed using the Immcantation framework (immcantation.org). V(D)J gene assignment on data obtained from *Nielsen et al., 2019* was performed using IgBLAST v1.13 (*Ye et al., 2013*) against the IMGT human germline reference database (*Giudicelli et al., 2005*) (IMGT/GENE-DB v3.1.24; retrieved August 3, 2019). V(D)J gene assignments and clonal cluster assignments were already available in all other non-OAS datasets and were retained. Nonproductively rearranged sequences were excluded. Using Change-O v1.0.0 (*Gupta et al., 2015*), the V and J genes of unmutated germline ancestors for each sequence were constructed with D segment and N/P regions masked by 'N' nucleotides. Sequence chimeras were filtered by removing any sequence with more than six mutations in any 10 nucleotide window. Individual immunoglobulin genotypes were computationally inferred using TIgGER v1.0.0 and used to finalize V(D)J annotations (*Gadala-Maria et al., 2015*). To infer clonal clusters, sequences were first partitioned based on common V and J gene annotations, and junction region length. Within these groups, sequences differing from one another by a specified Hamming distance threshold within the junction region were clustered into clones using single linkage hierarchical clustering (*Gupta et al., 2017*). The Hamming distance threshold was determined by finding

the local minimum of a bimodal distance to nearest sequence neighbor plot using SHazaM v1.0.2.999 (*Yaari et al., 2013*). In cases where automated threshold detection failed, usually because the distance to nearest neighbor distribution was not bimodal, the threshold was set to 0.1 and verified by manual inspection to ensure that a threshold of 0.1 was near a local minimum. Finally, the V and J genes of unmutated germline ancestors for each clone were constructed. Within these unmutated ancestral sequences, D segments and N/P regions were masked using ambiguous 'N' nucleotides.

## Testing for measurable evolution

Testing for measurable evolution begins with building B cell lineage trees. Within each B cell clone, identical sequences or those differing only by ambiguous nucleotides were collapsed unless they were sampled at different timepoints. To reduce computational complexity, lineages were randomly down-sampled to at most 500 sequences each. B cell lineage tree topologies and branch lengths were estimated using maximum parsimony using the pratchet function of the R package *phangorn* v2.5.5 (*Schliep, 2011*). R packages *dowser* v0.0.3 (*Hoehn et al., 2020*), *alakazam* v1.0.2.999 (*Gupta et al., 2015*), and *ape* v5.4-1 (*Paradis et al., 2004*) were used for phylogenetic analysis. Trees were visualized using *ggtree* v2.4.2 (*Yu et al., 2016*), and other figures were generated using *ggplot2* v3.3.5 (*Wickham, 2016*) and *ggpubr* v0.4.0 (*Kassambara, 2020*). R v3.6.1 (*R Development Core Team, 2017*) was used for analysis of measurable evolution except for data from *Davis et al., 2019*. Due to technical upgrades, figure generation and selection analysis were performed using R v4.0.3, as well as *ape* v5.5, *phangorn* v2.7.1, *shazam* v1.1.0, *alakazam* v1.1.0, and *dowser* v0.1.0. Data from *Davis et al., 2019* were also analyzed using these updated packages.

To test for measurable evolution over time, we use a modified version of the previously described phylogenetic date randomization test (*Duchêne et al., 2015*; *Murray et al., 2016*) implemented in *dowser* v0.0.3 (*Hoehn et al., 2020*). Briefly, for a given tree the divergence of each tip was calculated as the sum of branch lengths leading to the tree's most recent common ancestor (MRCA). Only branches directly between a tip and the tree's most recent common ancestor were used to calculate divergence. We next calculated the Pearson's correlation between the divergence and sampling time of each tip. Measurably evolving lineages should show a positive correlation between divergence and time (*Figure 1A*). Divergence from the lineage's predicted unmutated ancestral sequence rather than the MRCA could also be used. Because all sequences relate to the unmutated ancestral sequence through the MRCA node, this would add a constant additional divergence to all sequences, resulting in the same correlation as when the MRCA is used. We next identified monophyletic clades containing only sequences from a single timepoint (here referred to as 'clusters'). We then randomly permuted sampling times among clusters, such that all sequences within each cluster had the same, randomly chosen timepoint. We next measured the correlation between divergence and time in this randomized tree, and repeated the process 100,000 times. We then estimated the p value that the observed correlation between divergence and time was no greater than expected from random distribution of times among clusters. This p value was calculated as the proportion of permutation replicates that had an equal or higher correlation than in the observed tree. We used a pseudocount of one for this calculation. The minimum possible p value for a lineage was calculated as one divided by possible number of distinct cluster permutations.

We modified the date randomization test to account for the high degree of topological uncertainty of many B cell lineage trees. More specifically, B cell lineage trees often contain large clusters of zero-length branches (soft polytomies) that represent high uncertainty in branching order (e.g., *Figure 1— figure supplement 1*). In bulk BCR data, these polytomies may be due to PCR error or sequencing error. If polytomies are resolved randomly into bifurcations, this can produce more single-timepoint monophyletic clades than necessary and lead to a high false positive rate of the date randomization test (*Figure 1—figure supplements 1 and 2*). To ensure this source of uncertainty did not increase the false positive rate of our analyses, we resolved bifurcations within each polytomy such that sequences from the same timepoint were grouped into the fewest possible number of single-timepoint mono-phyletic clades before performing permutations. While we do not have direct evidence that polytomies in B cell lineages trees are produced from PCR error, the fact that resolving them reduces the rate of lineages with a significant negative correlation between divergence and time (a biologically implausible result, *Figure 1—figure supplement 2*) suggests they are at least in part due to technical artifacts.

The clustered date randomization approach is more conservative than tests that permute tips uniformly (e.g., *Unterman et al., 2020*), but has been shown to be less biased if different subpopulations are sampled at each timepoint (*Murray et al., 2016*). To explore the effect of this modeling choice, we repeated the analyses in *Table 1* using two-tailed clustered and uniform date randomization tests (*Figure 1—figure supplement 2*). Two-tailed tests can identify lineages with a significant positive or negative correlation between divergence and time. This is useful because a significant negative correlation between divergence and time is biologically implausible and represents a likely false positive result. Due to multiple testing under an alpha value of 0.025, we expect no more than 2.5% of lineages to have a significant negative correlation from these two-tailed tests. We found the uniform permutation test had a high rate of negatively evolving lineages (mean = 8.3%), indicating a high false positive rate. By contrast, the clustered permutation test without resolved polytomies had a mean rate of only 2.2% negatively evolving lineages, approximately as expected given an alpha value of 0.025. Resolving polytomies and then performing the clustered permutation test improved performance even more, with a mean rate of 1.2% negatively evolving lineages and no dataset having more than 2.8% of lineages negatively evolving. This analysis shows the uniform date randomization test is prone to false positives in empirical B cell data, while the clustered date randomization test with resolved polytomies corrects this issue. All other tests performed in this study used a one-tailed, clustered date randomization test with resolved polytomies and an alpha value of 0.05.

To identify and characterize measurably evolving lineages while adjusting for multiple testing, all lineages tested were pooled together and p values were adjusted using the Benjamini–Hochberg procedure (*Benjamini and Hochberg, 1995*) implemented in the function p.adjust (*R Development Core Team, 2017*). Lineages with adjusted p values <0.1 were referred to as adjusted measurably evolving lineages (*Figure 2B*). To determine whether lineages were measurably increasing in amino acid divergence (*Figure 3A*), we repeated the clustered date randomization test for each tree. However, instead of calculating divergence as the sum of phylogenetic branch lengths leading from each tip to the most recent common ancestor of the lineage, we calculated divergence as the number of nonambiguous amino acid differences between each tip and the lineage's clonal germline. The clustered permutation test then proceeded as before, using the same cluster assignments as in the nucleotide-based test. This tested whether sequences at later timepoints had more amino acid substitutions compared to the germline than sequences at earlier timepoints.

It is possible that the results reported are affected by the size (number of sequences) of lineages in each dataset. A large number of lineages without adequate power could result in a spurious lack of measurable evolution. To ensure the lineages included in each study were adequately powered, we included only lineages with at least 15 sequences, that were sampled over at least 3 weeks, and had a minimum possible p value <0.05 based on the number of distinct permutations of timepoints among clusters. If measurable evolution were still strongly confounded by lineage size even after these filtering steps, we would expect measurably evolving lineages to be larger on average than nonmeasurably evolving lineages. By contrast, measurably evolving lineages were significantly larger than nonmeasurably evolving lineages in only 5/21 datasets surveyed (*Figure 3—figure supplement 2*), indicating our results are not strongly confounded by lineage size.

## Inclusion of experimentally validated mAbs

To identify B cell lineages that likely bind to the antigen under study, we included experimentally validated monoclonal antibody (mAb) heavy chain sequences provided from multiple studies. This included multiple anti-HIV mAbs: 11 from *Liao et al., 2013*, 12 from *Doria-Rose et al., 2014*, 7 from *Johnson et al., 2018*, 42 from *Landais et al., 2017*, 31 from *Wu et al., 2015*, and 4 from *Huang et al., 2016*. *Doria-Rose et al., 2014* and *Wu et al., 2015* also provided 680 and 1033 bulk BCR sequences, respectively, identified as clonally related to the provided anti-HIV broadly neutralizing mAbs. These sequences were also included in processing and clonal clustering but were not labeled as experimentally validated mAbs. *Davis et al., 2019* provided 885 mAb heavy chain sequences, some of which were tested for binding against Ebola virus proteins. All of these sequences were included in processing and clonal clustering, but only 368 validated by ELISA to bind to Ebola virus were labeled as EBV-binding mAbs. All of the above sequences were processed in the same manner as bulk sequences from OAS, except they were not filtered as potential PCR chimeras. Clonal lineages containing experimentally validated mAbs were labeled as antigen-binding; however, because sample

timepoints were not always apparent, mAb sequences themselves were removed before lineage tree inference for the abovementioned studies. Processed data from *Turner et al., 2020* also included 196 anti-influenza mAbs. These sequences were retained during tree inference because they were explicitly labeled by timepoint and usually cloned from previously identified sequences within the data. Of all mAbs included, only the 58 clonally clustered within powered lineages (at least 15 sequences sampled over 3 weeks, and minimum p value <0.05) were included in tests of mAb enrichment (*Figure 2—figure supplement 3*).

## Simulation-based power analysis

We used simulations to determine whether the clustered date randomization test was sufficiently powered to detect ongoing B cell evolution. These analyses used the *bcr-phylo* package accessed 9/21/2020 (*Davidsen and Matsen, 2018*; *Ralph and Matsen, 2020*), which simulates clonal lineages of B cells undergoing affinity maturation against a target sequence. For all simulations, a random naive heavy chain sequence was chosen from those provided in *bcr-phylo* and the rate of SHM was set to the default of $\lambda = 0.356$, which corresponds to an SHM rate of ~0.001 SHM/site/division (*Teng and Papavasiliou, 2007*). Mutations were introduced according to the S5F model (*Yaari et al., 2013*). Selection strength was chosen to be either 0 (neutral) or 1 (entirely affinity driven). A single target sequence was chosen for affinity maturation. All other parameters were set to their default.

We performed two sets of simulations. In the first, we simulated single B cell lineages from which 50 cells were sampled at generation 10, and 50 more cells were sampled after a specified number of additional generations (*Figure 1G*, *Figure 1—figure supplement 3*). In the second type of simulation, we replicated the sampling strategy of *Laserson et al., 2014*. Briefly, for each clone in subject *hu420143* from *Laserson et al., 2014*, we simulated one lineage with the same number of cells (if enough cells had been generated) sampled after the number of generations corresponding to 1, 3, 7, 14, 21, and 28 days (*Figure 1—figure supplement 4*). The number of generations corresponding to each sample day was calculated using a strict generation time of either 12 or 24 hr, which are conservative given previous GC cycle estimates of 6–12 hr (*Allen et al., 2007*; *Hauser et al., 2007*; *Victora and Nussenzweig, 2012*). These simulations used a selection strength of 1, which gave more conservative results in previous simulations (*Figure 1—figure supplement 3*).

To account for possible issues with clonal clustering, we did not preserve clonal identities among simulated sequences in either simulation type. Instead, we pooled sequences from all simulation repetitions under a particular parameter set and used the same clonal clustering method used for empirical data analyses to group them into clonal clusters. We did not repeat the genotyping or chimera filtering steps done on empirical data analyses as genotyped individuals and sequence chimeras were not part of the simulations. We performed the clustered date randomization test with resolved polytomies on each lineage with a minimum possible p value <0.05. Because all sequences were simulated under affinity maturation, the proportion of lineages with $p < 0.05$ indicated the true positive rate of the test. To determine the false positive rate, we randomized sample times among tips within each tree and repeated the date randomization test (*Figure 1—figure supplements 3 and 4*). Here, the proportion with $p < 0.05$ indicated the false positive rate.

## Analysis of selection

To understand the force of selection operating on B cell lineages, we first separated all adjusted measurably evolving lineages into their respective subjects within each study. We then excluded all subjects with only one measurably evolving lineage. While all sequences included were labeled as productive by IgBlast, three contained premature stop codons in their IMGT-aligned sequences, likely due to insertions that were removed during alignment. These sequences were removed. For computational efficiency, all lineages were down-sampled to a maximum size of 100 sequences. Due to uncertainty in germline D-region assignment, only V-gene (IMGT positions 1–312) nucleotides were included for analyses of selection, similar to *Hoehn et al., 2017*. We then estimated lineage tree topologies, branch lengths, and subject-wide substitution model parameters under the GY94 model (*Hoehn et al., 2019*; *Nielsen and Yang, 1998*). Using fixed tree topologies estimated from the GY94 model, we then estimated branch lengths, subject-wide $\omega$ values for CDR and FWR partitions ($\omega_{CDR}$ and $\omega_{FWR}$), and all six canonical SHM hot- and cold-spot motif parameters under the HLP19 model in IgPhyML v1.1.3 (*Hoehn et al., 2019*) for all adjusted measurably evolving lineages. Significance of $\omega$

estimates was determined using two phylogenetic likelihood ratio tests, similar to *Hoehn et al., 2017*. To determine the significance of $\omega_{CDR}$ estimates, we compared the maximum log-likelihood obtained when both $\omega_{CDR}$ and $\omega_{FWR}$ were estimated by maximum likelihood ($L$) to that obtained when $\omega_{FWR}$ was estimated by maximum likelihood but $\omega_{CDR}$ was fixed at 1 ($L_{CDR=1}$). The likelihood ratio statistic (LRS) for this test was calculated as $2 \times (L - L_{CDR=1})$. Because these models differ by one freely estimated parameter, the LRS will be approximately chi-squared distributed with one degree of freedom under the null hypothesis that $\omega_{CDR} = 1$, which allows for p value calculation (*Huelsenbeck and Rannala, 1997*). To determine significance of $\omega_{FWR}$ estimates, the process is the same except LRS $= 2 \times (L - L_{FWR=1})$, where $L_{FWR=1}$ is the maximum log-likelihood obtained when $\omega_{CDR}$ was estimated by maximum likelihood but $\omega_{FWR}$ was fixed at 1 ($L_{FWR=1}$). All of the above statistics are reported in *Table 2*.

## Data and material availability

All data are publically available from prior publications. Script to reproduce all analyses performed are available at https://bitbucket.org/kleinstein/projects.git (*Kleinstein Lab, 2021*; copy archived at swh:1:rev:1ca83cda5d1baac880c71c314b0adc359314f6fa).

## Acknowledgements

We would like to thank Dr Louis Du Plessis for helpful discussion, and Dr Julian Q Zhou for providing processed data. This work was funded in part by National Institutes of Health, National Institute of Allergy and Infectious Diseases grant R01 AI104739, and by the European Research Council under the European Union's Seventh Framework Programme (FP7/2007-2013)/European Research Council grant agreement number 614725-PATHPHYLODYN. The Ellebedy laboratory was supported by NIAID grants R21 AI139813, U01 AI141990, and NIAID Centers of Excellence for Influenza Research and Surveillance (CEIRS) contract HHSN272201400006C to AHE. JST was supported by NIAID 5T32CA009547.

## Additional information

### Competing interests

Kenneth B Hoehn: receives consulting fees from Prellis Biologics. Jackson S Turner: is the recipient of a licensing agreement with Abbvie and has received consulting fees from Gerson Lehman Group. Ali H Ellebedy: The Ellebedy laboratory received funding under sponsored research agreements from Emergent BioSolutions and AbbVie. Steven H Kleinstein: receives consulting fees from Northrop Grumman and Peraton. The other authors declare that no competing interests exist.

### Funding

| Funder | Grant reference number | Author |
| --- | --- | --- |
| National Institute of Allergy and Infectious Diseases | R01 AI104739 | Steven H Kleinstein |
| European Research Council | 614725-PATHPHYLODYN | Oliver G Pybus |
| National Institute of Allergy and Infectious Diseases | R21 AI139813 | Dr Ali Ellebedy |
| National Institute of Allergy and Infectious Diseases | U01 AI141990 | Dr Ali Ellebedy |
| National Institute of Allergy and Infectious Diseases | HHSN272201400006C | Dr Ali Ellebedy |
| National Institute of Allergy and Infectious Diseases | 5T32CA009547 | Jackson S Turner |

The funders had no role in study design, data collection, and interpretation, or the decision to submit the work for publication.

## Author contributions
Kenneth B Hoehn, Conceptualization, Data curation, Formal analysis, Methodology, Software, Writing – original draft, Writing – review and editing; Jackson S Turner, Ruoyi Jiang, Data curation, Resources, Writing – review and editing; Frederick I Miller, Data curation, Software, Writing – review and editing; Oliver G Pybus, Methodology, Writing – review and editing; Ali H Ellebedy, Resources, Supervision, Writing – review and editing; Steven H Kleinstein, Conceptualization, Funding acquisition, Supervision, Writing – original draft, Writing – review and editing

## Author ORCIDs
Kenneth B Hoehn ⓘ http://orcid.org/0000-0003-0411-4307
Steven H Kleinstein ⓘ http://orcid.org/0000-0003-4957-1544

## Decision letter and Author response
Decision letter https://doi.org/10.7554/eLife.70873.sa1
Author response https://doi.org/10.7554/eLife.70873.sa2

# Additional files

## Supplementary files
• Transparent reporting form

## Data availability
The manuscript is a computational study. All data used are publicaly available. Source code are available at https://bitbucket.org/kleinstein/projects (copy archived at https://archive.softwareheritage.org/swh:1:rev:1ca83cda5d1baac880c71c314b0adc359314f6fa). All of the OAS datasets are available at this URL: http://opig.stats.ox.ac.uk/webapps/oas.

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
