## [Editor Report]

The manuscript by Hoehn et al., introduces a novel approach to measure evolution in B cell responses, and apply it to a wide variety of data sets. The work provides significant new insight into which stimuli induce effective immune responses, and which has the potential to improve vaccine design. This will be of interest to those interested in B cell responses, especially in the case of vaccinations that induce poor immune responses.

---

## [Decision Letter]

**Decision letter after peer review:**

Thank you for submitting your article "Human B cell lineages associated with germinal centers following influenza vaccination are measurably evolving" for consideration by *eLife*. Your article has been reviewed by 3 peer reviewers, and the evaluation has been overseen by a Reviewing Editor and Aleksandra Walczak as the Senior Editor. The reviewers have opted to remain anonymous.

Essential revisions:

1. The authors should try to either show (with simulations) that the data rejects the following scenario or they should include it as an equally likely possibility:

"Suppose that B cell clones expanded and diversified through somatic hypermutation prior to the study period (that is, prior to the secondary vaccination event which is the focus of the study). It seems that preferential expansion of highly mutated subclones during the study period could bias detected sequences towards more divergent sequences, even without ongoing somatic mutation during the study period. Preferential expansion of divergent sequences would give rise to higher average divergence as the study period goes on, giving the appearance of accumulation of additional mutations, but in fact these mutations had occurred prior to the study period and are simply more readily detected in the sparsely sampled repertoire sequencing data after their expansion. Far from being simply a pathological counter-example, this scenario seems biologically plausible, given that B cells harboring more divergent, affinity-matured sequences should generally have higher affinity antibodies that allow them to better compete for limited antigen and thus provide stronger division stimulus. This model predicts that some highly divergent sequences exist at early timepoints and would occasionally be detected."

2. Do you see any evidence for positive (or negative) selection during somatic evolution? Do you see any significant statistical difference between non-synonymous and synonymous SHMs?

3. Please expand the discussion on multiple testing adjustments in the section "Measurably evolving lineages following influenza vaccination show signs of memory B cell origin". Notably, the reported percentages of measurably evolving lineages in several scenarios (7.2% for primary hepatitis B vaccination; 6.5% for allergen-specific immunotherapy; 5.9% for HIV infection) are near the false positive rate of the test (5% of lineages measurably evolving). The authors have performed this test on datasets from ~21 studies, raising a concern that multiple hypothesis testing could give rise to false positives in some of the datasets. These results are interpreted as evidence of measurable evolution, even though they could seemingly be explained by the false discovery rate combined with multiple hypothesis testing. The authors should clarify how these results can be interpreted in light of the false positive rate of their test and multiple hypothesis testing, and must consider whether more conservative conclusions are warranted in these scenarios.

4. Did non-measurably evolving lineages also contain monoclonal antibodies that bound to vaccine antigens? Is there enrichment of vaccine-binding monoclonal antibodies within measurably evolving lineages?

5. Please modify the abstract in how the following two scenarios are presented: "some lineages enter GCs and thus likely undergo SHM", and "the average SHM over all lineages doesn't increase more than some threshold". These two scenarios are not contradictory and could both be true.

6. Figure 3A is very visually striking, despite the small sample sizes. Could you explain why this figure show a much stronger correlation than Figure S7.

7. p23 l22: why is the divergence from mrca (rather than naive ancestor) the one that we want for these tests?

8. p24 l15: do you have direct evidence for how much of the polytomy prevalence is from PCR/sequencing error?

9. The distribution of p-values should be plotted for all datasets as in Figure S6. It would be instructive to look at them and compare the full distribution given the two choices of significance threshold at 1% and 5%.

*Reviewer #1 (Recommendations for the authors):*

I have a number of questions about how or why different steps were undertaken, but none of them seem likely to significantly affect the basic conclusions.

– Abstract: I don't think the two findings are contradictory. To my understanding, the first says "some lineages enter GCs and thus likely undergo SHM", whereas the second says "the average SHM over all lineages doesn't increase more than some threshold". Since I think the first doesn't measure what "some" is, and since flu is usually given to non-naive individuals whose responses vary greatly depending on exposure history, and since the threshold could be too small to detect some SHM that occurs, both of these results seem compatible with what I would imagine is most researchers' prior: some lineages undergo SHM in some circumstances. The (in my view quite large) contribution of the current paper is in illuminating what both of the "some"s in the previous sentence mean. I think setting it up as a conflict between two prior results that (unless I'm misunderstanding) aren't actually in conflict just confuses the reader. As it says at p11 l21: "consistent with a primarily GC-independent memory B cell response and/or rarity of antigen-specific lineages in the peripheral blood". I prefer the framing in the first sentence of Discussion: "The extent to which seasonal influenza vaccination stimulates affinity maturation…"

– p2 l10: saying that you "demonstrate measurable evolution" in some cases seems like you care only about false negatives, but not false positives. I prefer the way this is framed at end of intro, as a "survey" with "significant heterogeneity" that conforms to expectations in both directions.

– p3 l10: "at a rate orders of magnitude".

– p3 l14: "and, rarely, re-enter".

– p3 l21: same comment as abstract: not convinced they're in conflict.

– p6 l10: "and a second time after the".

– Figure 2A:

– two categories of HIV here (empty/first 60 weeks) should match those in B (early/late).

– why is *hep* B not broken into naive/boost here? I assume T=0 is prime, T~1.4 is boost?

– suggest "healthy children" rather than "healthy" so reader can guess whether they expect enrichment for ME.

– Figure 2B: having ~half the y axis devoted to p values and "late hiv" makes it hard to compare everything else. I think main message is e.g. flu is like *hep* boost, but not *hep* naive, which i can only really tell by parsing the tiny p values at the top.

– p11 l21: my understanding is that equating "GC-independent" with "no SHM" isn't correct, e.g. this https://pubmed.ncbi.nlm.nih.gov/33326765/ takes as settled that some shm takes place outside the GC. Also, I could be wrong but it would make more sense to me to say something like "only a small fraction of existing flu lineages are restimulated" (which as you say earlier is relatively rare) as the first alternative.

– p14 l1 I would think that "occur at low frequency in the blood" might be better than "are not enriched in the blood", since the latter (to me) sounds like the bottleneck is only on exiting GCs, rather than the other (previous) steps.

– Figure 3A is very striking/convincing (although I guess given the small sample sizes almost worrisome it's so straight?). But could you explain why just by eye it seems so much more striking (stronger correlation) than Figure S7? I realize "min GC %" and "proportion GC B cells" are different, as are "-log P" and "% ME", and one is a scatter plot with low transparency and lots of dots are on top of each other, but 3A looks like almost a perfect relationship, whereas S7 it's hard to even see a linear relationship.

– p17 l19: this paragraph is great, it's really convincing to me.

– p19 l21: couldn't there also be a lot of lineages that are condition-specific and GC-derived, but not re-stimulated by the current stimulus? I don't know a number for the frequency with which re-stimulation causes an antigen-specific lineage to re-enter GCs, but I wouldn't expect it to be very close to 1.

– p21 l1: "to cover as wide a variety of conditions as possible".

– p21 l15: does "redundancy" mean number of observed sequences? Or does it have something to do with the number of nucleotide changes you could make without affecting the AA seq? (I presume the former, i just haven't heard it used in this way). Could use "multiplicity" or "observations" if they would be equivalent.

– p23 l2: what does "manual inspection" consist of? i.e. how do you know by eye that 0.1 is correct for non-bimodal distributions?

– p23 l3: What does "masking" consist of? Does this mean that you're not inferring the D/insert portions of the naive ancestral sequence?

– p23 l22: why is the divergence from mrca (rather than naive ancestor) the one that we want for these tests? Maybe the trunk bit would maybe just cancel out? But then at p26 l6 it looks like for the AA version you do compare to naive/germline seq? From intro to Duchene 2015, it seems they used root (not mrca)?

– p24 l15: do you have direct evidence for how much of the polytomy prevalence is from PCR/sequencing error? For instance do you get fewer polytomies in data with barcodes/UMIs?

– Figure S3: why is it so much easier to detect measurable evolution when we're looking at neutral evolution, i.e. what causes the long downward tails of points in the top right plot (selection strength 1) vs the left plot (neutral)?

– Figure S5: why does it look like there's only a lower bound/quantile (no upper box) for red (Boost, Standard)?

– I would find it very interesting if you could expand on the alternative explanations in the last paragraph of the Discussion. Partly because "does not result from a complete lack of vaccine-induced B cell evolution" seems like a very low bar/unlikely null hypothesis (i don't think many people thought there was zero).

– It might be worth discussing why you don't (I think) attempt to measure selection (it's fine that you didn't). You do an amino acid-based analysis, which is related to this (but doesn't discuss selection strength), and do simulations with both neutral and strong selection, but I'm curious why you focused only on detecting SHM/evolution, and not on whether it was neutral or not.

- It would also be nice to discuss why using parsimony (very heuristic, not very accurate) was preferred over more sophisticated methods.

*Reviewer #2 (Recommendations for the authors):*

Congratulations on the paper! I enjoyed reading the preprint, and I only have a few comments and suggestions that I list below.

1. For more clarity, the distribution of p-values should be plotted for all datasets as in Figure S6. It would be instructive to look at them and compare the full distribution given the two choices of significance threshold at 1% and 5%. To this end, I think it would make sense to plot the cumulative function.

2. I think the section "Measurably evolving lineages following influenza vaccination show signs of memory B cell origin" could use a more extensive explanation of the multiple testing adjustment. The p-values distributions would also be important here to distinguish the standard randomization test p-values with the BH adjusted p-values. Detecting lineages using the second definition of the p-value should also be tested with synthetic datasets.

3. The initial germline divergence is quantified using the sum of branch lengths for each lineage. I suppose this depends strongly on the lineage size (and that one on the experimental protocol). Is there a way to control for this? (For instance, would it make sense to look at these distributions for subsampled lineages of equal size?)

4. In the discussion you refer to the rates of somatic hypermutation and the length of the GC cycles as given by the literature you cite. For completeness (perhaps as a supplementary figure), could you report the values of the slope fitted in the SHM number vs sample time plot for measurably evolving lineages (as in Figure 1B)? I would be curious to see how these numbers compare with independent estimations from the data and whether their distribution changes significantly between cohorts you've studied.

5. Re discussion on page 11: Even if memory B cells do not re-enter GC, one could imagine detecting the ongoing evolution of naive cells – this possibility should be discussed. Later the results suggest the evolving lineages come mainly from memory cells (page 12) but a priori both scenarios could be true.

6. Figure 1G misses the y axis label and the x axis label is somewhat confusing without reference to the main text. The fractions in boxes should be written with the "%" sign (also in other figures).

7. In Figure 2A the point corresponding to the early-childhood dataset should be distinguishable from other healthy data (I guess it's the "significant" green point).

The caption of Figure 2B should use the term "initial germline divergence" again, as in the y axis label to avoid confusion.

8. Page 5 line 3: before using "SHM/site" first, it would be better to say what it means in words.

Page 5 line 4: In evolving lineages, sequences *sampled* at later time points are (…).

---

## [Author Response]

Essential revisions:1. The authors should try to either show (with simulations) that the data rejects the following scenario or they should include it as an equally likely possibility:"Suppose that B cell clones expanded and diversified through somatic hypermutation prior to the study period (that is, prior to the secondary vaccination event which is the focus of the study). It seems that preferential expansion of highly mutated subclones during the study period could bias detected sequences towards more divergent sequences, even without ongoing somatic mutation during the study period. Preferential expansion of divergent sequences would give rise to higher average divergence as the study period goes on, giving the appearance of accumulation of additional mutations, but in fact these mutations had occurred prior to the study period and are simply more readily detected in the sparsely sampled repertoire sequencing data after their expansion. Far from being simply a pathological counter-example, this scenario seems biologically plausible, given that B cells harboring more divergent, affinity-matured sequences should generally have higher affinity antibodies that allow them to better compete for limited antigen and thus provide stronger division stimulus. This model predicts that some highly divergent sequences exist at early timepoints and would occasionally be detected."

We agree that this is a potential alternative explanation for a significant positive correlation between divergence and time, and have now addressed this as a possibility in the Results and Discussion sections. We have also removed explicit references to detecting “ongoing SHM” in the text, in favor terms that more directly reflect what our test detects such as “B cell evolution” or “increasing SHM frequency” which do not imply novel SHM over the sampling interval. Nevertheless, we believe our results are more easily explained as a result of ongoing SHM, and have added some text making that point. In the context of influenza vaccination, day 5 plasmablasts represent the breadth of the B cell memory pool. If measurable evolution were due solely from preferential recall, we would expect the divergences of sequences at later timepoints to fall within the range of day 5 plasmablasts. Instead, in the high-GC influenza binding lineages we identified (Figure 3B/C), many late-sampled GC sequences are clearly more diverged from the day 5 plasmablast response. Further, if measurable evolution from influenza vaccination were due simply to preferential re-stimulation of highly mutated B cells, we would expect influenza binding lineages without any GC sequences to be measurably evolving. To test this, we repeated the analysis in Figure 3A using only lineages containing influenza-binding monoclonal antibodies (mAbs). Results were highly consistent with Figure 3A: influenza-binding lineages without GC sequences were less likely to be evolving than those with high proportions of GC sequences **(**Figure 3 —figure supplement 3). Thus, significant GC involvement, rather than simply binding to influenza, is more predictive of measurable evolution. All of these points are more easily explained if measurable evolution is the result of additional SHM. Nonetheless, we cannot definitely rule out this alternative explanation, we have highlighted both possible mechanisms of B cell evolution. We have included descriptions of this new analysis in the Results (pp. 14-15) and Discussion (pp. 20-21).

2. Do you see any evidence for positive (or negative) selection during somatic evolution? Do you see any significant statistical difference between non-synonymous and synonymous SHMs?

We have now included additional analysis of selection in a new section in the Results (pp. 1113), Methods (pp. 32-33) and in Table 2. We quantified the ratio of non-synonymous to synonymous substitutions (dN/dS) for CDRs and FWRs for all adjusted (and non-adjusted) measurably evolving lineages. We found evidence of purifying selection (dN/dS < 1), which is explainable as part of normal affinity maturation, and consistent with previous observations (Hoehn et al., 2019). We further added a discussion of these results to the Discussion section (p. 17).

3. Please expand the discussion on multiple testing adjustments in the section “Measurably evolving lineages following influenza vaccination show signs of memory B cell origin”. Notably, the reported percentages of measurably evolving lineages in several scenarios (7.2% for primary hepatitis B vaccination; 6.5% for allergen-specific immunotherapy; 5.9% for HIV infection) are near the false positive rate of the test (5% of lineages measurably evolving). The authors have performed this test on datasets from ~21 studies, raising a concern that multiple hypothesis testing could give rise to false positives in some of the datasets. These results are interpreted as evidence of measurable evolution, even though they could seemingly be explained by the false discovery rate combined with multiple hypothesis testing. The authors should clarify how these results can be interpreted in light of the false positive rate of their test and multiple hypothesis testing, and must consider whether more conservative conclusions are warranted in these scenarios.

We appreciate the reviewer’s concern and have added a new section on multiple hypothesis testing to the Discussion detailing these caveats (pp. 19-20), as well as additional details to the relevant Results sections (p. 10). We also repeated our initial germline divergence analysis that used “adjusted” measurably evolving lineages without the multiple testing correction, and found similar results (Figure 2 —figure supplement 4). We also repeated these analyses using a more strict cutoff (adjusted p < 0.05), which also yielded similar results (Figure 2 —figure supplement 4). These are discussed in the main text on p. 11.

4. Did non-measurably evolving lineages also contain monoclonal antibodies that bound to vaccine antigens? Is there enrichment of vaccine-binding monoclonal antibodies within measurably evolving lineages?

Yes, both measurably evolving and non-measurably evolving lineages contained antigen-binding mAbs. We added a new analysis to the Results (p. 9) showing that measurably evolving lineages were enriched for mAbs in nearly all studies that included mAbs relevant to the condition being studied (Figure 2 —figure supplement 3). This was also true in our analysis of influenza vaccination. While intriguing, we also detail multiple caveats for this new analysis in the Discussion (p. 18), including the low numbers of mAbs in individual studies and the fact that mAbs are not always randomly chosen among lineages. We also added further detail about how mAbs were selected and included in the Methods section (p. 30).

5. Please modify the abstract in how the following two scenarios are presented: “some lineages enter GCs and thus likely undergo SHM”, and “the average SHM over all lineages doesn’t increase more than some threshold”. These two scenarios are not contradictory and could both be true.

We have rephrased this sentence in the abstract to state both claims without implying they necessarily contradict each other. We also removed a sentence explicitly stating these results are contradictory in the Introduction (p. 3).

6. Figure 3A is very visually striking, despite the small sample sizes. Could you explain why this figure show a much stronger correlation than Figure S7.

These figures are showing different metrics. Figure 3A shows the minimum GC % on the x axis, while Figure S7 (now Figure 3 —figure supplement 1) shows the raw GC %. We have changed Figure 3 —figure supplement 1 to show the correlation between divergence and time rather than the -log10(p value). We believe this shows the relationship much more clearly. We have also included a bootstrap analysis of this relationship to demonstrate its significance. These are detailed in the main text on p. 13.

7. P23 l22: why is the divergence from mrca (rather than I ancestor) the one that we want for these tests?

Divergence from the lineage’s predicted germline ancestor sequence rather than the MRCA could also be used. However, because all sequences relate to the germline sequence through the MRCA node, this would add a constant additional divergence to all sequences, resulting in the same correlation as when the MRCA is used. We have now made note of this in the Methods (p. 26).

8. P24 l15: do you have direct evidence for how much of the polytomy prevalence is from PCR/sequencing error?

No, but circumstantially this seems like a real possibility. Further, without polytomy resolution we see biologically implausible patterns such as significant negative evolution in some lineages (Figure 1 —figure supplement 2). Because of this we believe this is an important factor to account for. We have now made note of this in the Results (p. 5) Methods (p. 27) and Discussion (p. 20).

9. The distribution of p-values should be plotted for all datasets as in Figure S6. It would be instructive to look at them and compare the full distribution given the two choices of significance threshold at 1% and 5%.

We have now included these in the revised Figure 2 —figure supplement 1, referenced in the main text on p. 10.

Reviewer #1 (Recommendations for the authors):I have a number of questions about how or why different steps were undertaken, but none of them seem likely to significantly affect the basic conclusions.– abstract: I don’t think the two findings are contradictory. To my understanding, the first says “some lineages enter GCs and thus likely undergo SHM”, whereas the second says “the average SHM over all lineages doesn’t increase more than some threshold”. Since I think the first doesn’t measure what “some” is, and since flu is usually given to non-I individuals whose responses vary greatly depending on exposure history, and since the threshold could be too small to detect some SHM that occurs, both of these results seem compatible with what I would imagine is most researchers’ prior: some lineages undergo SHM in some circumstances. The (in my view quite large) contribution of the current paper is in illuminating what both of the “some”s in the previous sentence mean. I think setting it up as a conflict between two prior results that (unless I’m misunderstanding) aren’t actually in conflict just confuses the reader. As it says at p11 l21: “consistent with a primarily GC-independent memory B cell response and/or rarity of antigen-specific lineages in the peripheral blood”. I prefer the framing in the first sentence of Discussion: “The extent to which seasonal influenza vaccination stimulates affinity maturation…”

As detailed in Essential Revisions, we have edited the abstract so that these two concepts are not framed as in conflict.

– p2 l10: saying that you “demonstrate measurable evolution” in some cases seems like you care only about false negatives, but not false positives. I prefer the way this is framed at end of intro, as a “survey” with “significant heterogeneity” that conforms to expectations in both directions.

We have edited this in the abstract.

– p3 l10: “at a rate orders of magnitude”.– p3 l14: “and, rarely, re-enter”.– p3 l21: same comment as abstract: not convinced they’re in conflict.– p6 l10: “and a second time after the”.

Thanks for catching these typos! We have fixed them. We also changed “and, rarely, re-enter” to read “or possibly re-enter” which more accurately reflects the potential fate of memory B cells.

– Figure 2A:– two categories of HIV here (empty/first 60 weeks) should match those in B (early/late).

These are meant to be different categories. In Figure 2A, the “first 60 weeks” are derived from all HIV studies and only include sequences from the first 60 weeks of sampling. The early/late categories in 2B refer to how long after HIV infection the study was performed. These were chosen to emphasize different aspects of the data. In Figure 2A we wanted to show that the sample interval did not determine the extent of evolution (thus subsampled data to 60 weeks). In Figure 2B we separated based on the time since infection because we were interested in initial divergence at the first timepoint. We’ve clarified this in the caption for Figure 2B. It is also worth noting that there are only 5 “60 week” HIV studies rather than 8 because three studies (Huang et al., 2016, Schanz et al., 2014, and Wu et al., 2015) did not have two sample points within the first year of sampling, and were thus not included in “HIV, first 60 weeks.” We have made note of this in the Figure 2 caption.

– why is hep B not broken into naive/boost here? I assume T=0 is prime, T~1.4 is boost?– suggest "healthy children" rather than "healthy" so reader can guess whether they expect enrichment for ME.

We have made both of these changes in the updated Figure 2 caption.

– Figure 2B: having ~half the y axis devoted to p values and "late hiv" makes it hard to compare everything else. I think main message is e.g. flu is like hep boost, but not hep naive, which i can only really tell by parsing the tiny p values at the top.

We have increased the font size for the p values and lowered the bars somewhat for this figure.

The most interesting comparison for us was that between influenza vaccination and early/late HIV. This is because early vs late HIV likely represent early vs chronic primary immune response, while influenza vaccination likely represents a memory response. Because of that, we kept the HIV categories as close to the influenza results as possible.

– p11 l21: my understanding is that equating "GC-independent" with "no SHM" isn't correct, e.g. this https://pubmed.ncbi.nlm.nih.gov/33326765/ takes as settled that some shm takes place outside the GC. Also, I could be wrong but it would make more sense to me to say something like "only a small fraction of existing flu lineages are restimulated" (which as you say earlier is relatively rare) as the first alternative.

We agree with this reviewer that “GC independent” does not necessarily mean “no SHM” is occurring. However, in this sentence in the text we are making the converse claim, that “no SHM” indicates “GC independence.” We think this is a more defensible statement, since GCs are organs dedicated to SHM and affinity maturation (now p. 10).

– p14 l1 I would think that "occur at low frequency in the blood" might be better than "are not enriched in the blood", since the latter (to me) sounds like the bottleneck is only on exiting GCs, rather than the other (previous) steps.

We have changed this to “do not occur at high frequency in the blood” since the observed frequencies are close to the expected false positive rate (now p. 10).

– Figure 3A is very striking/convincing (although I guess given the small sample sizes almost worrisome it's so straight?). But could you explain why just by eye it seems so much more striking (stronger correlation) than Figure S7? I realize "min GC %" and "proportion GC B cells" are different, as are "-log P" and "% ME", and one is a scatter plot with low transparency and lots of dots are on top of each other, but 3A looks like almost a perfect relationship, whereas S7 it's hard to even see a linear relationship.

We have addressed this in the Essential Revisions.

– p17 l19: this paragraph is great, it's really convincing to me.

Thanks!

– p19 l21: couldn't there also be a lot of lineages that are condition-specific and GC-derived, but not re-stimulated by the current stimulus? I don't know a number for the frequency with which re-stimulation causes an antigen-specific lineage to re-enter GCs, but I wouldn't expect it to be very close to 1.

We have now revised the text to address this as a possibility (now p. 22).

– p21 l1: "to cover as wide a variety of conditions as possible".

We have made this change (p. 23).

– p21 l15: does "redundancy" mean number of observed sequences? Or does it have something to do with the number of nucleotide changes you could make without affecting the AA seq? (I presume the former, i just haven't heard it used in this way). Could use "multiplicity" or "observations" if they would be equivalent.

Yes. We have changed this to “multiplicity” as recommended (p. 23).

– p23 l2: what does "manual inspection" consist of? i.e. how do you know by eye that 0.1 is correct for non-bimodal distributions?

We have added more detail to clarify the approach. Briefly, it means we looked at the distributions to verify 0.1 was not too far away from an identifiable minimum.

– p23 l3: What does "masking" consist of? Does this mean that you're not inferring the D/insert portions of the naive ancestral sequence?

Yes. We have now clarified this in the text (p. 25).

– p23 l22: why is the divergence from mrca (rather than naive ancestor) the one that we want for these tests? Maybe the trunk bit would maybe just cancel out? But then at p26 l6 it looks like for the AA version you do compare to naive/germline seq? From intro to Duchene 2015, it seems they used root (not mrca)?

Addressed in Essential Revisions.

– p24 l15: do you have direct evidence for how much of the polytomy prevalence is from PCR/sequencing error? For instance do you get fewer polytomies in data with barcodes/UMIs?

Addressed in Essential Revisions.

– Figure S3: why is it so much easier to detect measurable evolution when we're looking at neutral evolution, i.e. what causes the long downward tails of points in the top right plot (selection strength 1) vs the left plot (neutral)?

We suspect this is due to selection reducing the rate of divergence in the simulations, therefore lowering the power of the test. We confirmed that simulations that include selection had lower average germline divergence than neutral simulations (Figure 1 —figure supplement 5). This makes sense, as selection can only reduce genetic diversity. We’ve addressed this now on p. 6.

– Figure S5: why does it look like there's only a lower bound/quantile (no upper box) for red (Boost, Standard)?

The lower bound whisker was covered by two of the dots. We re-ran these analyses which, in addition to changing the numerical results slightly also moved the dots out of the way (Figure 2 —figure supplement 2).

– I would find it very interesting if you could expand on the alternative explanations in the last paragraph of the Discussion. Partly because "does not result from a complete lack of vaccine-induced B cell evolution" seems like a very low bar/unlikely null hypothesis (i don't think many people thought there was zero).

We appreciate the reviewer’s interest in this question. This is an area of active investigation for us, and we are obtaining new data to look into alternative explanations. However, for this paper we wanted to avoid speculating too much on what could be driving the low efficacy of influenza vaccination. We’ve made it clearer that future work will be needed to test these alternative explanations (p. 22).

– It might be worth discussing why you don't (I think) attempt to measure selection (it's fine that you didn't). You do an amino acid-based analysis, which is related to this (but doesn't discuss selection strength), and do simulations with both neutral and strong selection, but I'm curious why you focused only on detecting SHM/evolution, and not on whether it was neutral or not.

We have addressed this in the Essential Revisions.

- It would also be nice to discuss why using parsimony (very heuristic, not very accurate) was preferred over more sophisticated methods.

We have added an explanation to the Discussion (p. 18).

Reviewer #2 (Recommendations for the authors):Congratulations on the paper! I enjoyed reading the preprint, and I only have a few comments and suggestions that I list below.

Thanks! Glad you liked the paper.

1. For more clarity, the distribution of p-values should be plotted for all datasets as in Figure S6. It would be instructive to look at them and compare the full distribution given the two choices of significance threshold at 1% and 5%. To this end, I think it would make sense to plot the cumulative function.

We have addressed this in the Essential Revisions.

2. I think the section "Measurably evolving lineages following influenza vaccination show signs of memory B cell origin" could use a more extensive explanation of the multiple testing adjustment. The p-values distributions would also be important here to distinguish the standard randomization test p-values with the BH adjusted p-values. Detecting lineages using the second definition of the p-value should also be tested with synthetic datasets.

We have addressed this in the Essential Revisions.

3. The initial germline divergence is quantified using the sum of branch lengths for each lineage. I suppose this depends strongly on the lineage size (and that one on the experimental protocol). Is there a way to control for this? (For instance, would it make sense to look at these distributions for subsampled lineages of equal size?)

The divergence is not the sum of all branch lengths within a lineage, which is more appropriately referred to as the “diversity” of a lineage. Rather, the divergence for an individual tip is calculated by tracing the branches from that tip to the most recent common ancestor (MRCA). The divergence is the sum of only those branches between the tip and the MRCA. In principle, it should not be strongly affected by the lineage size. An example is shown in Author response image 1, with a full tree at top and a subsampled tree at bottom. The red branches show the lengths that are added up to get the divergence for the selected tip, and are the same despite so many tips being dropped. While having fewer sequences may move the MRCA node, ultimately that should not affect the analysis because it is only the relative divergence at each timepoint that is relevant for the correlation test. We note that in practice long phylogenetic branch lengths tend to be underestimated with maximum parsimony methods, so the divergence of a tip can often increase when trees are sampled more completely. However, this is a much more minor effect than the difference between diversity and divergence. We have now added text to clarify the definition of divergence in the Methods section. While we expect statistical power to detect measurable evolution to be generally higher in larger lineages, we show in Figure 3 —figure supplement 2 that measurably evolving lineages are significantly larger in only 5/21 datasets surveyed, indicating lineage size is not enough to drive a trend towards measurable evolution.

**Author response image 1. sa2fig1:** 

4. In the discussion you refer to the rates of somatic hypermutation and the length of the GC cycles as given by the literature you cite. For completeness (perhaps as a supplementary figure), could you report the values of the slope fitted in the SHM number vs sample time plot for measurably evolving lineages (as in Figure 1B)? I would be curious to see how these numbers compare with independent estimations from the data and whether their distribution changes significantly between cohorts you've studied.

We thank the reviewer for this suggestion, and agree that the results could make for an interesting comparison. However, we do not believe that this analysis is directly related to the claims we are making in this manuscript. It is certainly an area we are looking into for future work, however.

5. Re discussion on page 11: Even if memory B cells do not re-enter GC, one could imagine detecting the ongoing evolution of naive cells – this possibility should be discussed. Later the results suggest the evolving lineages come mainly from memory cells (page 12) but a priori both scenarios could be true.

We have now added text to address this possibility in the Results (p. 9).

6. Figure 1G misses the y axis label and the x axis label is somewhat confusing without reference to the main text. The fractions in boxes should be written with the "%" sign (also in other figures).

We added the y-axis label to Figure 1 and the % symbol to Figure 1 and relevant supplemental figures.

7. In Figure 2A the point corresponding to the early-childhood dataset should be distinguishable from other healthy data (I guess it's the "significant" green point).The caption of Figure 2B should use the term "initial germline divergence" again, as in the y axis label to avoid confusion.

We have made these changes to distinguish healthy adults and children in Figure 2 and Table 1.

8. Page 5 line 3: before using "SHM/site" first, it would be better to say what it means in words.Page 5 line 4: In evolving lineages, sequences sampled at later time points are (…).

We have made these changes.